# A G$_s$-coupled purinergic receptor boosts Ca$^{2+}$ influx and vascular contractility during diabetic hyperglycemia

**Maria Paz Prada[1†], Arsalan U Syed[1†], Olivia R Buonarati[1†], Gopireddy R Reddy[1], Matthew A Nystoriak[2], Debapriya Ghosh[1], Sergi Simó[3], Daisuke Sato[1], Kent C Sasse[4], Sean M Ward[5], Luis F Santana[6], Yang K Xiang[1,7], Johannes W Hell[1], Madeline Nieves-Cintrón[1]\*, Manuel F Navedo[1]\***

[1]Department of Pharmacology, University of California, Davis, Davis, United States; [2]Diabetes & Obesity Center, Department of Medicine, University of Louisville, Kentucky, United States; [3]Department of Cell Biology & Human Anatomy, University of California, Davis, Davis, United States; [4]Sasse Surgical Associates, Reno, United States; [5]Department of Physiology & Cell Biology, University of Nevada, Reno, United States; [6]Department of Physiology & Membrane Biology, University of California, Davis, Davis, United States; [7]VA Northern California Healthcare System, Mather, United States

**\*For correspondence:**
mcnieves@ucdavis.edu (MN-Có);
mfnavedo@ucdavis.edu (MFN)

[†]These authors contributed equally to this work

**Competing interests:** The authors declare that no competing interests exist.

**Abstract** Elevated glucose increases vascular reactivity by promoting L-type Ca$_V$1.2 channel (LTCC) activity by protein kinase A (PKA). Yet, how glucose activates PKA is unknown. We hypothesized that a G$_s$-coupled P2Y receptor is an upstream activator of PKA mediating LTCC potentiation during diabetic hyperglycemia. Experiments in apyrase-treated cells suggested involvement of a P2Y receptor underlying the glucose effects on LTTCs. Using human tissue, expression for P2Y$_{11}$, the only G$_s$-coupled P2Y receptor, was detected in nanometer proximity to Ca$_V$1.2 and PKA. FRET-based experiments revealed that the selective P2Y$_{11}$ agonist NF546 and elevated glucose stimulate cAMP production resulting in enhanced PKA-dependent LTCC activity. These changes were blocked by the selective P2Y$_{11}$ inhibitor NF340. Comparable results were observed in mouse tissue, suggesting that a P2Y$_{11}$-like receptor is mediating the glucose response in these cells. These findings established a key role for P2Y$_{11}$ in regulating PKA-dependent LTCC function and vascular reactivity during diabetic hyperglycemia.
DOI: https://doi.org/10.7554/eLife.42214.001

## Introduction

Diabetes is a major risk factor contributing to cardiovascular complications, including hypertension, increased risk of stroke, coronary disease, and organ failure (*Whelton et al., 2018*). These complications are often linked to the effects of elevated extracellular glucose (e.g. hyperglycemia - the hallmark metabolic abnormality in diabetes) on vascular function (*Brown et al., 2010*; *Cooper et al., 2001*; *Tousoulis et al., 2013*). In addition to endothelium-dependent alterations (*Lagaud et al., 2001*; *Sena et al., 2013*), endothelium-independent modifications, including enhanced contractility of arterial myocytes, are emerging as critical contributing factors to these complications (*Aronson, 2008*; *DeFronzo et al., 2013*; *Fleischhacker et al., 1999*; *Montero et al., 2013*; *Nieves-Cintrón et al., 2018*). Consistent with this notion, we and others recently demonstrated augmented vasoconstriction in response to acute and chronic elevations in extracellular glucose that were independent of endothelial function (*Fleischhacker et al., 1999*; *Jackson et al., 2016*; *Nystoriak et al.,*

*2017*). We linked these glucose-mediated effects on vascular reactivity to an increase in L-type $Ca_V1.2$ channel (LTCC) activity in human and murine arterial myocytes (*Navedo et al., 2010b*; *Nystoriak et al., 2017*). This is critical, as $Ca^{2+}$ influx via LTCCs is the main $Ca^{2+}$ entry pathway in these cells and determines muscle contractility and arterial diameter (*Knot and Nelson, 1998*). Unexpectedly, we found that glucose-mediated potentiation of LTCCs involves phosphorylation of $Ca_V1.2$ at serine 1928 ($Ser^{1928}$) by protein kinase A (PKA) (*Morotti et al., 2017*; *Navedo et al., 2010b*; *Nystoriak et al., 2017*). Yet, how glucose leads to PKA activation is unknown.

Elevations in cellular glucose have been shown to trigger autocrine release of nucleotides (e.g. ATP, UTP) into the extracellular space of many cells, including arterial myocytes (*Costa et al., 2009*; *Hazama et al., 1998*; *Nilsson et al., 2006*; *Parodi et al., 2002*). These extracellular nucleotides can then act on G protein-coupled P2Y receptors to induce a plethora of cellular responses (*von Kügelgen and Harden, 2011*). Previous studies demonstrated that elevations in extracellular glucose led to increases in intracellular $Ca^{2+}$ ($[Ca^{2+}]_i$) in isolated arterial myocytes (*Nilsson et al., 2006*). Indeed, this glucose-mediated response was associated with enhanced autocrine release of nucleotides engaging P2Y receptors, but the contributing mechanisms as well as the identity of the P2Y receptor involved are unclear.

Eight functionally diverse G protein-coupled P2Y receptors have been identified (*von Kügelgen and Harden, 2011*). Most of these P2Y receptors are coupled to $G_{q/i}$ proteins. Conversely, $P2Y_{11}$ is the only P2Y receptor that is coupled to $G_s$ (*Communi et al., 1997*; *Kennedy, 2017*; *Schuchardt et al., 2012*; *von Kügelgen and Harden, 2011*). Activation of this receptor could then activate adenylyl cyclase (AC) and PKA to underlie the glucose-induced effects on LTCC activity, but its role in arterial myocytes is unclear. $P2Y_{11}$ receptors were first cloned from human tissue and subsequently identified in other species (*Kennedy, 2017*; *von Kügelgen and Harden, 2011*). Although the $P2Y_{11}$ gene has not been found at the expected position in the mouse genome, recent rodent annotations related to this gene (e.g. XM_008766009.2 and XM_0130655917.2), as well as studies defining a functional role based on pharmacological data obtained from rat and mouse tissue hinted at the presence of at least a $P2Y_{11}$-like receptor in mice (*Kennedy, 2017*). We, therefore, undertook a comprehensive approach to investigate whether upstream engagement of this $G_s$-coupled P2Y receptor is implicated in PKA activation leading to increased $Ser^{1928}$ phosphorylation of vascular $Ca_V1.2$ that results in potentiation of LTCC activity during diabetic hyperglycemia. Accordingly, Western blot analysis, immunofluorescence confocal microscopy, Ground State Depletion (GSD) super-resolution nanoscopy, FRET-based cAMP imaging and voltage-clamp electrophysiology confirmed functional expression of $P2Y_{11}$ in human arterial myocytes. We found that elevated glucose stimulates cAMP synthesis with the same magnitude as that observed upon application of the selective $P2Y_{11}$ agonist NF546 or when both stimuli were given simultaneously, thus suggesting that glucose and NF546 signaling proceed through the same pathway. Nanometer proximity was observed between $P2Y_{11}$ and PKA and $P2Y_{11}$ and $Ca_V1.2$ at the sarcolemma of arterial myocytes, which facilitates a structural arrangement that may be essential for glucose effects on LTCC activity. Indeed, the NF546/glucose-induced cAMP signal resulted in enhanced $Ser^{1928}$ phosphorylation and LTCC activity in human arterial myocytes, and these changes were prevented by the selective $P2Y_{11}$ inhibitor NF340. Intriguingly, comparable functional data were observed in mouse arterial myocytes, suggesting that a $P2Y_{11}$-like receptor could be mediating the glucose response in these cells that results in vasoconstriction (*Kennedy, 2017*). Thus, our data in human and murine tissue indicate a key role for a $G_s$-coupled P2Y receptor, which fits the distribution, pharmacological, and signaling profile of a $P2Y_{11}$ or $P2Y_{11}$-like receptor, respectively, as an upstream player in the regulation of LTCCs and vascular reactivity during diabetic hyperglycemia. These findings may have important clinical and therapeutic implications as they may point to $P2Y_{11}$ as a potential target for treating diabetic vascular complications.

## Results

### Glucose-induced extracellular nucleotide release mediates $Ser^{1928}$ phosphorylation, increased LTCC activity and vasoconstriction

In our first series of experiments, we pressurized (60 mmHg) mouse cerebral arteries to develop stable and spontaneous tone (*Supplementary file 1* and *2*). When increasing extracellular glucose from

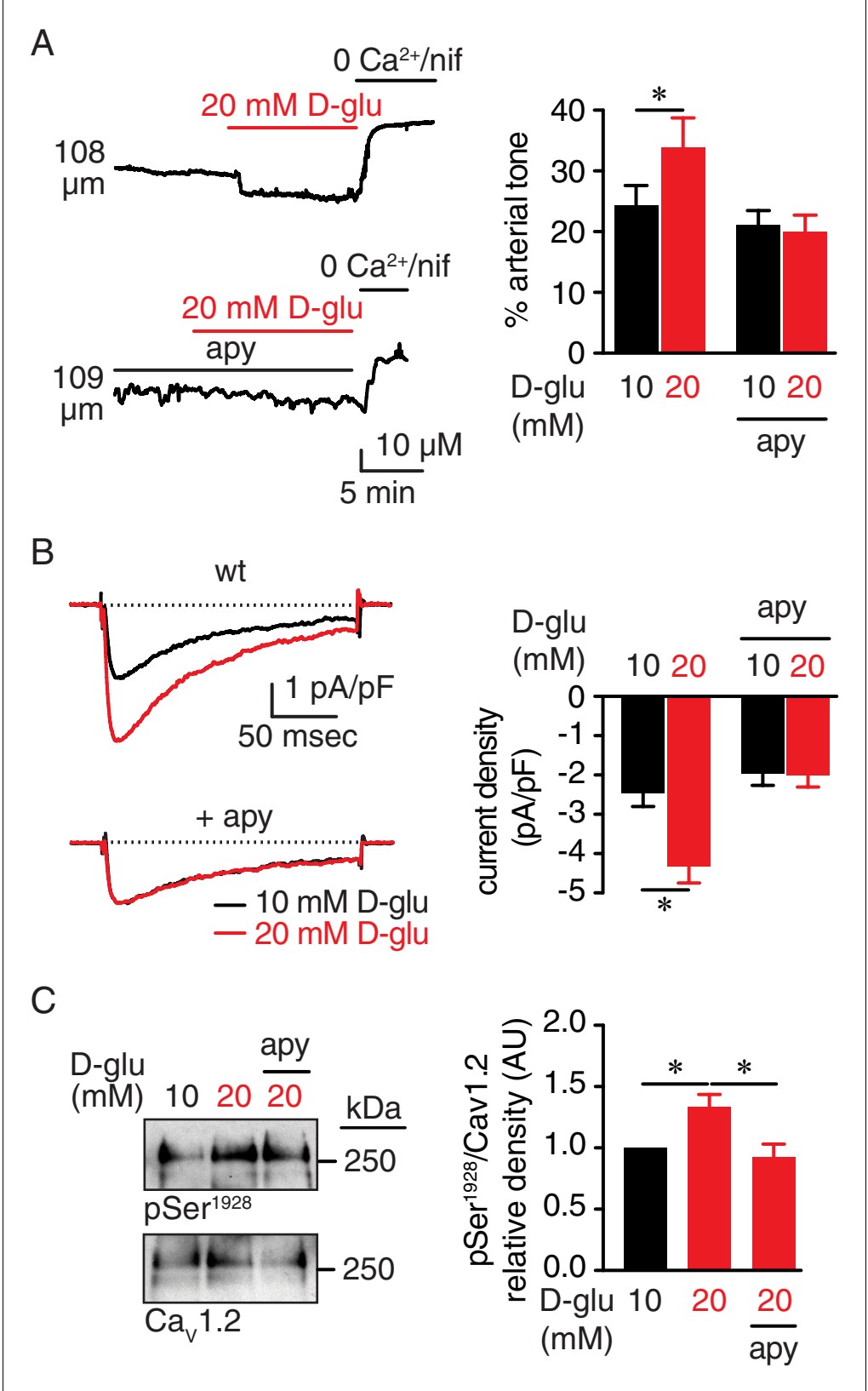

**Figure 1.** Extracellular nucleotides promote vasoconstriction, Ca$_V$1.2 activity and Ser[1928] phosphorylation in response to 20 mM D-glucose in murine arterial myocytes. (**A**) Representative diameter recordings and summary arterial tone data from pressurized (60 mmHg) wt mouse cerebral arteries before and after application of 20 mM

*Figure 1 continued on next page*

*Figure 1 continued*

D-glucose in the absence (n = 6 arteries from six mice) and presence (n = 6 arteries from six mice) of apyrase (apy; 0.32 U/ml; *p<0.05, Wilcoxon matched pairs test; *Figure 1—source data 1*). (B) Characteristic $I_{Ba}$ recordings from the same cell and summary $I_{Ba}$ data from wt mouse cerebral arterial myocytes evoked by step depolarizations from −70 to +10 mV before and after application of 20 mM D-glucose in the absence (n = 11 cells from five mice) and presence of apyrase (n = 9 cells from five mice) (*p<0.05, paired *t* test; *Figure 1—source data 2*). (C) Representative immunoblot detection of phosphorylated $Ser^{1928}$ ($pSer^{1928}$) and total $Ca_V1.2$ from wt mouse cerebral and mesenteric arteries after 10 min incubation with 10 mM or 20 mM D-glucose in the absence and presence of apyrase (n = 10 arterial lysates per condition), and quantification of $pSer^{1928}$ (AU = arbitrary units) (*p<0.05, Kruskal-Wallis with Dunn's multiple comparisons; *Figure 1—source data 3*).
DOI: https://doi.org/10.7554/eLife.42214.002

The following source data and figure supplements are available for figure 1:

**Source data 1.** Excel spreadsheet containing the individual numeric values of % arterial tone analyzed in *Figure 1A* and corresponding raw diameters.
DOI: https://doi.org/10.7554/eLife.42214.010

**Source data 2.** Excel spreadsheet containing the individual numeric values of current density analyzed in *Figure 1B*.
DOI: https://doi.org/10.7554/eLife.42214.011

**Source data 3.** Excel spreadsheet containing the individual numeric values of $pSer^{1928}$/$Ca_V1.2$ relative density analyzed in *Figure 1C*.
DOI: https://doi.org/10.7554/eLife.42214.012

**Figure supplement 1.** $K^+$-induced arterial constriction in the absence and presence of apyrase, no changes in arterial tone, $I_{Ba}$ and $pSer^{1928}$ in response to 20 mM mannitol, and full-length blots corresponding to *Figure 1C*.
DOI: https://doi.org/10.7554/eLife.42214.003

**Figure supplement 1—source data 1.** Excel spreadsheet containing the individual numeric values of 60 mM $K^+$-induced % constriction analyzed in *Figure 1—figure supplement 1A*.
DOI: https://doi.org/10.7554/eLife.42214.004

**Figure supplement 1—source data 2.** Excel spreadsheet containing the individual numeric values of % arterial tone analyzed in *Figure 1—figure supplement 1B* and corresponding raw diameters.
DOI: https://doi.org/10.7554/eLife.42214.005

**Figure supplement 1—source data 3.** Excel spreadsheet containing the individual numeric values of current density analyzed in *Figure 1—figure supplement 1C*.
DOI: https://doi.org/10.7554/eLife.42214.006

**Figure supplement 1—source data 4.** Excel spreadsheet containing the individual numeric values of $pSer^{1928}$/α-tubulin relative density analyzed in *Figure 1—figure supplement 1E*.
DOI: https://doi.org/10.7554/eLife.42214.007

**Figure supplement 2.** Enhanced $I_{Ba}$ in response to elevated glucose is prevented by continuous bath perfusion.
DOI: https://doi.org/10.7554/eLife.42214.008

**Figure supplement 2—source data 1.** Excel spreadsheet containing the individual numeric values of current density analyzed in *Figure 1—figure supplement 2B*.
DOI: https://doi.org/10.7554/eLife.42214.009

10 mM to 20 mM D-glucose, there was a significant constriction (*Figure 1A* and *Supplementary file 1* and *2*), which is consistent with previous reports by our group and others (*Jackson et al., 2016*; *Nystoriak et al., 2017*). These extracellular D-glucose concentrations are used because they are within the range of observed nonfasting glucose levels in nondiabetic and diabetic mouse models (*Lagaud et al., 2001*; *Moien-Afshari et al., 2008*; *Navedo et al., 2010b*; *Nieves-Cintrón et al., 2015*; *Nystoriak et al., 2014*; *Nystoriak et al., 2017*). Considering that elevating glucose in the external milieu may trigger autocrine release of nucleotides (*Costa et al., 2009*; *Hazama et al., 1998*; *Nilsson et al., 2006*; *Parodi et al., 2002*), we investigated whether hydrolysis of extracellular nucleotides would ablate glucose-mediated vasoconstriction. For these experiments, pressurized arteries (60 mmHg) were pre-treated with the ectonucleotidase apyrase (0.32 U/mL) in a 10 mM D-glucose solution for 10 min. Apyrase-treated arteries failed to constrict to 20 mM D-glucose (*Figure 1A* and *Supplementary file 2*). Vasoconstriction in response to 60 mM potassium ($K^+$) in which the membrane potential closely follows the equilibrium potential for $K^+$ (~−20 mV) (*Knot and Nelson, 1998*), was similar in control and apyrase-treated arteries (*Figure 1—figure supplement 1A* and *Supplementary file 1* and *2*). These results suggest that differences in glucose-induced

vasoconstriction were not due to an inability of apyrase-treated arteries to respond to 20 mM D-glucose. Furthermore, the glucose-induced constriction was not attributed to changes in osmolarity as treatment with equimolar concentrations of the nonpermeable mannitol (20 mM = 10 mM D-glucose +10 mM mannitol) had no effect on vascular reactivity (*Figure 1—figure supplement 1B* and *Supplementary file 1* and *2*).

LTCC activity is critical for vascular reactivity (*Knot and Nelson, 1998*), and its function is enhanced in response to acute increases in glucose and in diabetes due to increased Ser[1928] phosphorylation in the $Ca_V1.2$ carboxy terminal (*Morotti et al., 2017*; *Navedo et al., 2010b*; *Nystoriak et al., 2017*). We, thus, examined whether nucleotide degradation with apyrase prevents glucose-mediated increases in LTCC activity and Ser[1928] phosphorylation. To assess this, we used patch-clamp electrophysiology in the whole-cell perforated configuration with barium ($Ba^{2+}$) as the charge carrier, before and after application of nifedipine to determine the nifedipine-sensitive $Ba^{2+}$ current ($I_{Ba}$) associated with LTCC activity in control and apyrase-treated cerebral arterial myocytes. We found that 20 mM D-glucose significantly enhanced $I_{Ba}$ in wild type (wt) control arterial myocytes, but not in apyrase-treated cells (*Figure 1B*). The glucose effect on LTCC activity was not due to changes in osmolarity as 20 mM mannitol had no impact on $I_{Ba}$ (*Figure 1—figure supplement 1C*).

We next performed Western blot analysis with a specific phospho-antibody against Ser[1928] at $Ca_V1.2$ and a FP1 antibody for total $Ca_V1.2$ detection that have been extensively validated by our group (*Buonarati et al., 2017*; *Davare et al., 1999*; *Hall et al., 2013*; *Nystoriak et al., 2017*; *Patriarchi et al., 2016*). Indeed, we have previously shown that the immunoreactivity at 250 kDa of $Ca_V1.2$ with the phospho-specific antibody for Ser[1928] (but not a phospho-specific antibody for Ser[1700]) and FP1 are completely eliminated in tissue from S1928A knockin mice (*Patriarchi et al., 2016*) and conditional $Ca_V1.2$ knockout mice (*Buonarati et al., 2017*), respectively. Data revealed increased phosphorylation of this residue in response to 20 mM D-glucose in control arterial lysates (*Figure 1C* and *Figure 1—figure supplement 1D*). However, this response was completely absent in arterial lysates exposed to 20 mM mannitol, thus ruling out any osmolarity effects (*Figure 1—figure supplement 1E*). Moreover, glucose-induced Ser[1928] phosphorylation was hindered in lysates from apyrase-treated arteries (*Figure 1C* and *Figure 1—figure supplement 1D*). These results suggest that elevated glucose increases Ser[1928] phosphorylation to potentiate LTCC activity through an autocrine mechanism involving secreted nucleotides.

To further test this possibility, LTCC activity was recorded in response to elevated glucose during continuous perfusion or static bath conditions (as above). The rationale for these experiments was to determine whether the wash out of nucleotides with flow can impact the glucose-induced potentiation of LTCCs. We found statistically similar $I_{Ba}$ in cells exposed to 10 mM and 20 mM D-glucose under continuous bath perfusion (*Figure 1—figure supplement 2*). Conversely, $I_{Ba}$ was significantly elevated in arterial myocytes exposed to 20 mM D-glucose when the bath perfusion was stopped (e.g. static bath conditions). These results suggest that secreted nucleotides could diffuse away under continuous flow, whereas under static bath conditions, they could accumulate in or around the area of release at the surface membrane of arterial myocytes to induce the optimal activation of the signaling pathway underlying increased LTCC activity in response to elevations in extracellular glucose.

## $P2Y_{11}$ receptors are in close proximity to $Ca_V1.2$ and $PKA_{cat}$ in arterial myocytes

Extracellular nucleotides are well-known activators of G-coupled P2Y receptors (*von Kügelgen and Harden, 2011*). $P2Y_{11}$ is a $G_s$-coupled P2Y receptor that could activate PKA (*Communi et al., 1997*; *Kennedy, 2017*; *Schuchardt et al., 2012*; *von Kügelgen and Harden, 2011*), and this could lead to enhanced Ser[1928] phosphorylation, LTCC activity, and vasoconstriction during diabetic hyperglycemia. Using Western blot analysis, we first validated the $P2Y_{11}$ antibody in tsA-201 cells, which endogenously express $P2Y_{11}$ (*Dreisig and Kornum, 2016*), by over-expressing human GFP-tagged $P2Y_{11}$ and by knocking down the endogenous and over-expressed receptor with specific $P2Y_{11}$ antisense oligodeoxynucleotides (ODNs) (*Figure 2A and B*). We subsequently detected an immunoreactive band of the expected molecular weight for $P2Y_{11}$ (~40 kDa) in lysates from freshly dissected human adipose arteries (*Figure 2C* and *Figure 2—figure supplement 1A*). This band seems specific for $P2Y_{11}$ since treatment of human arteries with specific human $P2Y_{11}$ antisense ODNs knocked

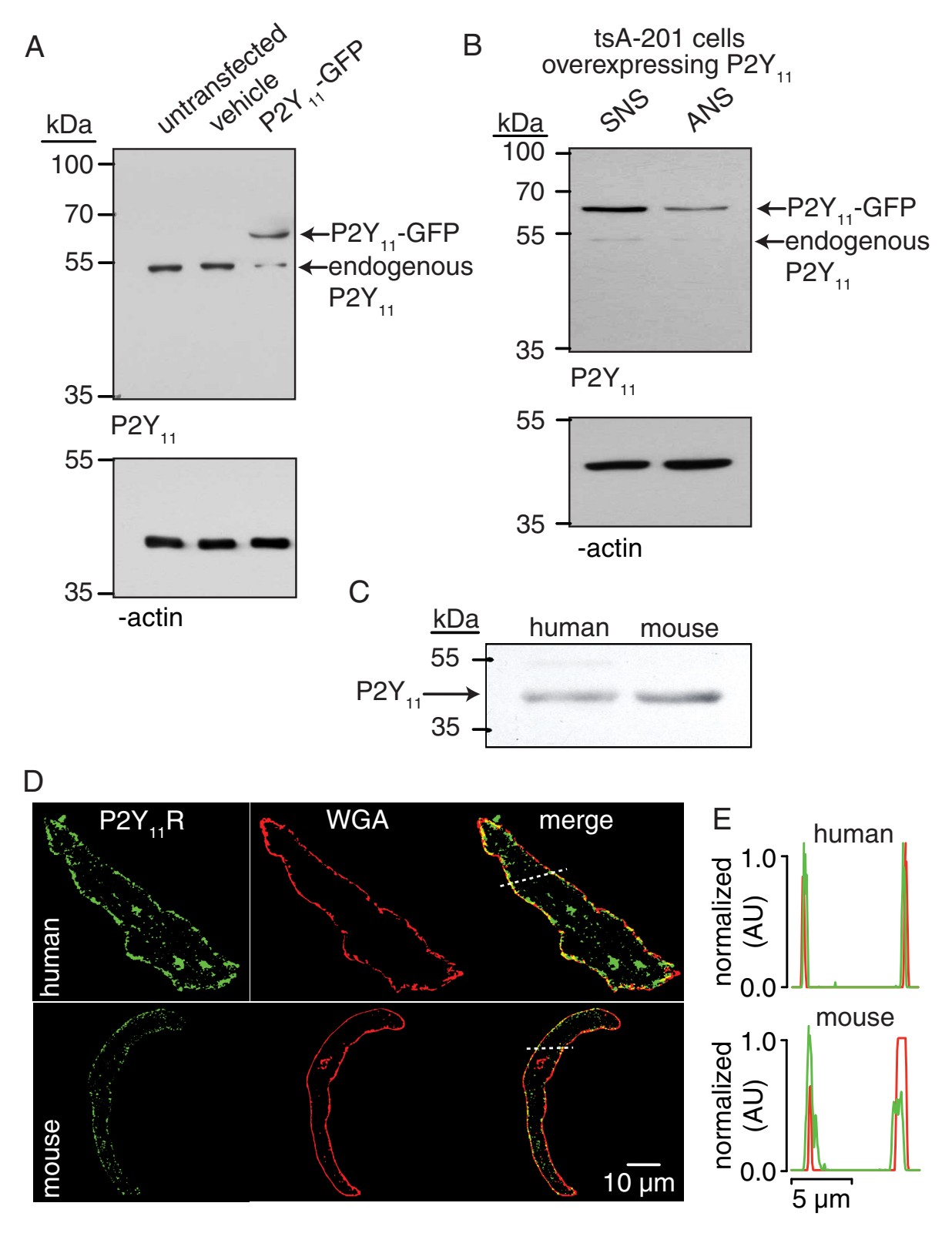

**Figure 2.** P2Y$_{11}$ protein and distribution in arterial myocytes. (**A**) Representative blot of immunoreactive bands of expected molecular weight for endogenous P2Y$_{11}$ (~40 kDa), overexpressed P2Y$_{11}$-GFP (~70 kDa), and β-actin (~43 kDa) in untransfected, vehicle-treated (empty transfection) and P2Y$_{11}$-GFP transfected tsA-201 cells (n = 3 lysates per condition). Note that tsA-201 cells endogenously express P2Y$_{11}$ (***Dreisig and Kornum, 2016***). (**B**) Representative blot of immunoreactive bands of expected molecular weight for endogenous P2Y$_{11}$ (~40 kDa), overexpressed P2Y$_{11}$-GFP (~70 kDa), and

*Figure 2 continued on next page*

*Figure 2 continued*

β-actin (~43 kDa) in tsA-cells transfected with P2Y$_{11}$-GFP as well as corresponding P2Y$_{11}$ sense (SNS) or antisense (ANS) ODNs (64% reduction in endogenous P2Y$_{11}$ expression in cells treated with ANS; 62% reduction in P2Y$_{11}$-GFP expression in P2Y$_{11}$-GFP-transfected cells treated with ANS; n = 3 lysates per condition; *Figure 2—source data 1*). (C) Representative immunoblot detection of P2Y$_{11}$ (~40 kDa) in lysates from human and wt mouse arteries (n = 3 arterial lysates per sample). (D) Representative confocal images of P2Y$_{11}$-associated fluorescence (green), wheat germ agglutinin (WGA, red) and merged channels in human (n = 11 cells from three humans) and wt mouse (n = 14 cells from three mice) arterial myocytes. (E) Line profile of the P2Y$_{11}$- and WGA-associated fluorescence from the area highlighted by the dotted lines in the representative human and mouse arterial myocytes in D.

DOI: https://doi.org/10.7554/eLife.42214.013

The following source data and figure supplements are available for figure 2:

**Source data 1.** Excel spreadsheet containing the individual numeric values of P2Y$_{11}$/ β-actin relative density corresponding to values reported in legend of *Figure 2B*.
DOI: https://doi.org/10.7554/eLife.42214.016

**Figure supplement 1.** Full-length blot for *Figure 2C*, knock down of P2Y$_{11}$ in arterial lysates, P2Y$_{11}$ immunoreactivity in isolated mouse arterial lysates, negative controls for immunofluorescence experiments in *Figure 2D* and P2Y$_{11}$ antibody control.
DOI: https://doi.org/10.7554/eLife.42214.014

**Figure supplement 1—source data 1.** Excel spreadsheet containing the individual numeric values of P2Y$_{11}$/ β-actin relative density analyzed in *Figure 2—figure supplement 1B*.
DOI: https://doi.org/10.7554/eLife.42214.015

down ~50% of the basal P2Y$_{11}$ protein abundance (*Figure 2—figure supplement 1B*). Confocal imaging and line profile analysis of P2Y$_{11}$-associated fluorescence and the plasma membrane marker wheat germ agglutinin (WGA) suggest a prominent distribution of the receptor on the surface membrane of freshly dissociated human arterial myocytes (*Figure 2D and E*). This was not observed when the antibody for P2Y$_{11}$ was omitted from the preparation (*Figure 2—figure supplement 1C*). These results suggest expression of P2Y$_{11}$ in human arterial myocytes. We observed comparable results in mouse arterial lysates and arterial myocytes, including detection and knock down with P2Y$_{11}$ ODNs of an immunoreactive band of the expected molecular weight for P2Y$_{11}$ (*Figure 2C* and *Figure 2—figure supplement 1A, B and D*). A band of ~110–120 kDa was observed in mouse arterial lysates, perhaps reflecting the formation of heterodimers between the P2Y$_{11}$ receptor itself or other purinergic/G protein-coupled receptors, as previously reported (*Barragán-Iglesias et al., 2015*; *Barragán-Iglesias et al., 2014*; *Ecke et al., 2008*; *Nishimura et al., 2016*). We also found expected distribution of P2Y$_{11}$-like-associated fluorescence at the surface membrane of mouse arterial myocytes (*Figure 2D and E*), which was not observed when the primary antibody was omitted or boiled (*Figure 2—figure supplement 1C and E*).

If P2Y$_{11}$ is involved in PKA-mediated activation of LTCCs in response to elevated glucose, we hypothesized that a subpopulation of these receptors should be localized in close proximity to Ca$_V$1.2 and PKA$_{cat}$. To test this possibility, we examined the spatial relationship of P2Y$_{11}$ with Ca$_V$1.2 and PKA$_{cat}$ in freshly dissociated human arterial myocytes using GSD super-resolution nanoscopy in the Total Internal Reflection Fluorescence (TIRF) configuration. This approach achieves a lateral resolution of ~20 nm with selective illumination of the (sub)sarcolemmal region. Note that the FP1 antibody against Ca$_V$1.2 has been extensively validated by our group (*Buonarati et al., 2017*; *Hall et al., 2013*; *Nystoriak et al., 2017*; *Patriarchi et al., 2016*). The PKA$_{cat}$ antibody was validated in arterial myocytes by pre-absorption with a blocking peptide (*Figure 3—figure supplement 1*). The GSD super-resolution localization maps (*Figure 3A and B*, *bottom panels*) for P2Y$_{11}$, Ca$_V$1.2, and PKA$_{cat}$ obtained from conventional TIRF images (*Figure 3A and B*, *top panels*) showed that these proteins form clusters of various sizes (*Figure 3C*) and density (*Figure 3D*) at the sarcolemma of human arterial myocytes. Whereas clusters were broadly expressed throughout the sarcolemma, merged spatial maps of P2Y$_{11}$ with Ca$_V$1.2 or PKA$_{cat}$ suggested a close association between a subset of these proteins (*Figure 3E and F*). Indeed, histograms of the P2Y$_{11}$-to-nearest Ca$_V$1.2 or PKA$_{cat}$ distances revealed that the closest centroids for P2Y$_{11}$-Ca$_V$1.2 and P2Y$_{11}$- PKA$_{cat}$ were at 59 nm and 58 nm, respectively (*Figure 3G and H*). Similar results were observed using freshly dissociated mouse cerebral arterial myocytes (*Figure 3—figure supplement 2*). Clusters of P2Y$_{11}$, Ca$_V$1.2, or PKA$_{cat}$ were never observed when primary antibodies were omitted from the human or mouse preparation (*Figure 3—figure supplement 3A and B*). To determine whether the interaction between a

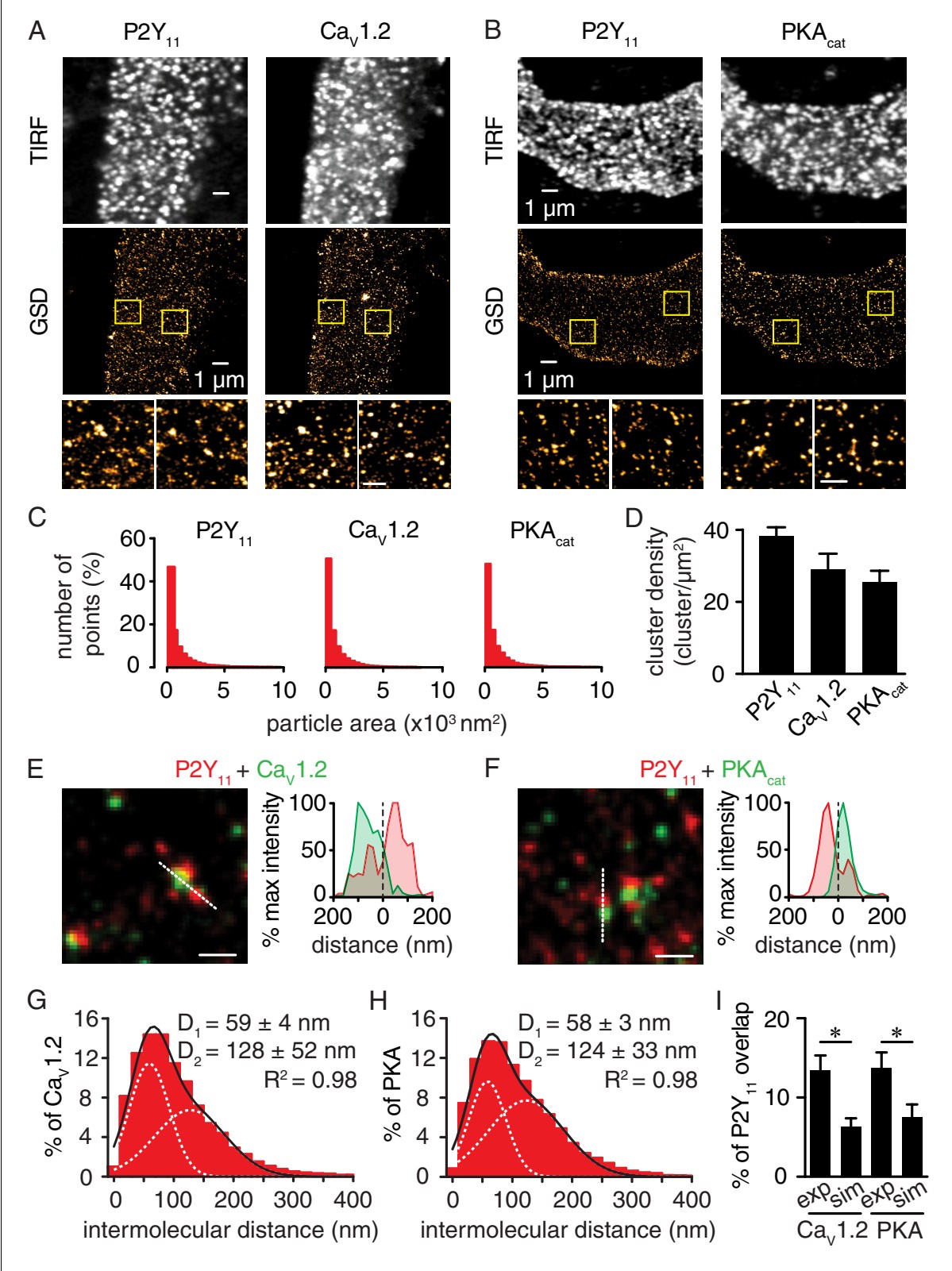

**Figure 3.** Nanometer organization of P2Y$_{11}$ with Ca$_V$1.2 and PKA$_{cat}$ in human arterial myocytes. Representative conventional TIRF images (top) and corresponding GSD reconstruction maps (bottom) from human arterial myocytes labeled for (A) P2Y$_{11}$ and Ca$_V$1.2 and (B) P2Y$_{11}$ and PKA$_{cat}$. Lower panels display enhanced magnifications of areas shown in yellow boxes (scale bar, 400 nm). (C) Histograms of the area of P2Y$_{11}$, Ca$_V$1.2 and PKA$_{cat}$ clusters in isolated human arterial myocytes (1621 ± 29, 1209 ± 16 and 1322 ± 20 nm$^2$, respectively; *Figure 3—source data 1*). (D) Bar plot of cluster

*Figure 3 continued*

density for P2Y$_{11}$, Ca$_V$1.2 and PKA$_{cat}$ (38 ± 2, 29 ± 4, and 25 ± 3 clusters/μm$^2$, respectively; *Figure 3—source data 2*). Enlarged merged image (*left*) and associated *x-y* fluorescence intensity profile (*right*) from area highlighted by the dotted lines of sites of close proximity between (**E**) P2Y$_{11}$ (red) and Ca$_V$1.2 (green) and (**F**) P2Y$_{11}$ (red) and PKA$_{cat}$ (green) (scale bar, 200 nm). Histograms of the lowest intermolecular distance to P2Y$_{11}$ centroids for (**G**) Ca$_V$1.2 (n = 19,611 particles from 6 cells; *Figure 3—source data 3*) and (**H**) PKA$_{cat}$ (n = 22,425 particles from 6 cells; *Figure 3—source data 4*) fluorescence particles. Data were fit with a sum of two Gaussian functions with depicted R$^2$ and centroids. (**I**) Bar plot of % overlap of P2Y$_{11}$ with Ca$_V$1.2 or PKA$_{cat}$ for experimental (Ca$_V$1.2: n = 36 segments from 12 cells; PKA$_{cat}$: n = 22 segments from 11 cells) and randomized simulations images (Ca$_V$1.2: n = 6 segments from 6 cells; PKA$_{cat}$: n = 6 segments from 6 cells) (*p<0.05, unpaired *t* test with Welch's correction; *Figure 3—source data 5*).
DOI: https://doi.org/10.7554/eLife.42214.017

The following source data and figure supplements are available for figure 3:

**Source data 1.** Excel spreadsheet containing the individual numeric values of frequency distribution histograms for cluster area in *Figure 3C*.
DOI: https://doi.org/10.7554/eLife.42214.025

**Source data 2.** Excel spreadsheet containing the individual numeric values for cluster density in *Figure 3D*.
DOI: https://doi.org/10.7554/eLife.42214.026

**Source data 3.** Excel spreadsheet containing the individual numeric values of frequency distribution histograms for intermolecular distance in *Figure 3G*.
DOI: https://doi.org/10.7554/eLife.42214.027

**Source data 4.** Excel spreadsheet containing the individual numeric values of frequency distribution histograms for intermolecular distance in *Figure 3H*.
DOI: https://doi.org/10.7554/eLife.42214.028

**Source data 5.** Excel spreadsheet containing the individual numeric values for % of P2Y$_{11}$ overlap in *Figure 3I*.
DOI: https://doi.org/10.7554/eLife.42214.029

**Figure supplement 1.** Validation for PKA$_{cat}$ primary antibody.
DOI: https://doi.org/10.7554/eLife.42214.018

**Figure supplement 2.** Nanometer organization of P2Y$_{11}$, Ca$_V$1.2 and PKA$_{cat}$ in murine arterial myocytes.
DOI: https://doi.org/10.7554/eLife.42214.019

**Figure supplement 2—source data 1.** Excel spreadsheet containing the individual numeric values of frequency distribution histograms for cluster area in *Figure 3—figure supplement 2C*.
DOI: https://doi.org/10.7554/eLife.42214.020

**Figure supplement 2—source data 2.** Excel spreadsheet containing the individual numeric values for cluster density in *Figure 3—figure supplement 2D*.
DOI: https://doi.org/10.7554/eLife.42214.021

**Figure supplement 2—source data 3.** Excel spreadsheet containing the individual numeric values of frequency distribution histograms for intermolecular distance in *Figure 3—figure supplement 2G*.
DOI: https://doi.org/10.7554/eLife.42214.022

**Figure supplement 2—source data 4.** Excel spreadsheet containing the individual numeric values of frequency distribution histograms for intermolecular distance in *Figure 3—figure supplement 2H*.
DOI: https://doi.org/10.7554/eLife.42214.023

**Figure supplement 3.** Negative controls for GSD images in human and murine arterial myocytes, and experimental and randomized reconstruction maps.
DOI: https://doi.org/10.7554/eLife.42214.024

subpopulation of P2Y$_{11}$ with a pool of Ca$_V$1.2 and PKA in arterial myocytes is the result of specific organization between two proteins, we compared the percentage of P2Y$_{11}$ and Ca$_V$1.2/PKA$_{cat}$ overlap between experimental super-resolution localization maps and images containing randomized distribution of P2Y$_{11}$, Ca$_V$1.2 and PKA$_{cat}$. Randomized images for P2Y$_{11}$-Ca$_V$1.2 and P2Y$_{11}$-PKA$_{cat}$ were generated based on data derived from the experimental super-resolution localization maps for these pairs of proteins using the Coste's randomization algorithm included in the JACoP plug-in in ImageJ (*Bolte and Cordelières, 2006*). Our analysis found that the percentage of overlap between P2Y$_{11}$-Ca$_V$1.2 and P2Y$_{11}$-PKA$_{cat}$ obtained from the experimental localization maps was significantly higher than that observed for a simulated random distribution between P2Y$_{11}$-Ca$_V$1.2 and P2Y$_{11}$-PKA$_{cat}$ (*Figure 3I* and *Figure 3—figure supplement 3C*). These results suggest an intimate association between subpopulations of P2Y$_{11}$-Ca$_V$1.2 and P2Y$_{11}$-PKA$_{cat}$.

We used the proximity ligation assay (PLA) as an additional analytical tool to examine the association between P2Y$_{11}$ and Ca$_V$1.2 or PKA$_{cat}$. PLA fluorescence puncta are only observed if proteins of interest are 40 nm or less apart (*Fredriksson et al., 2002*). We have extensively validated this approach in previous studies (*Nieves-Cintrón et al., 2017*; *Nystoriak et al., 2017*). Whereas PLA signals were absent when at least one primary antibody was omitted (*Figure 4—figure supplement*

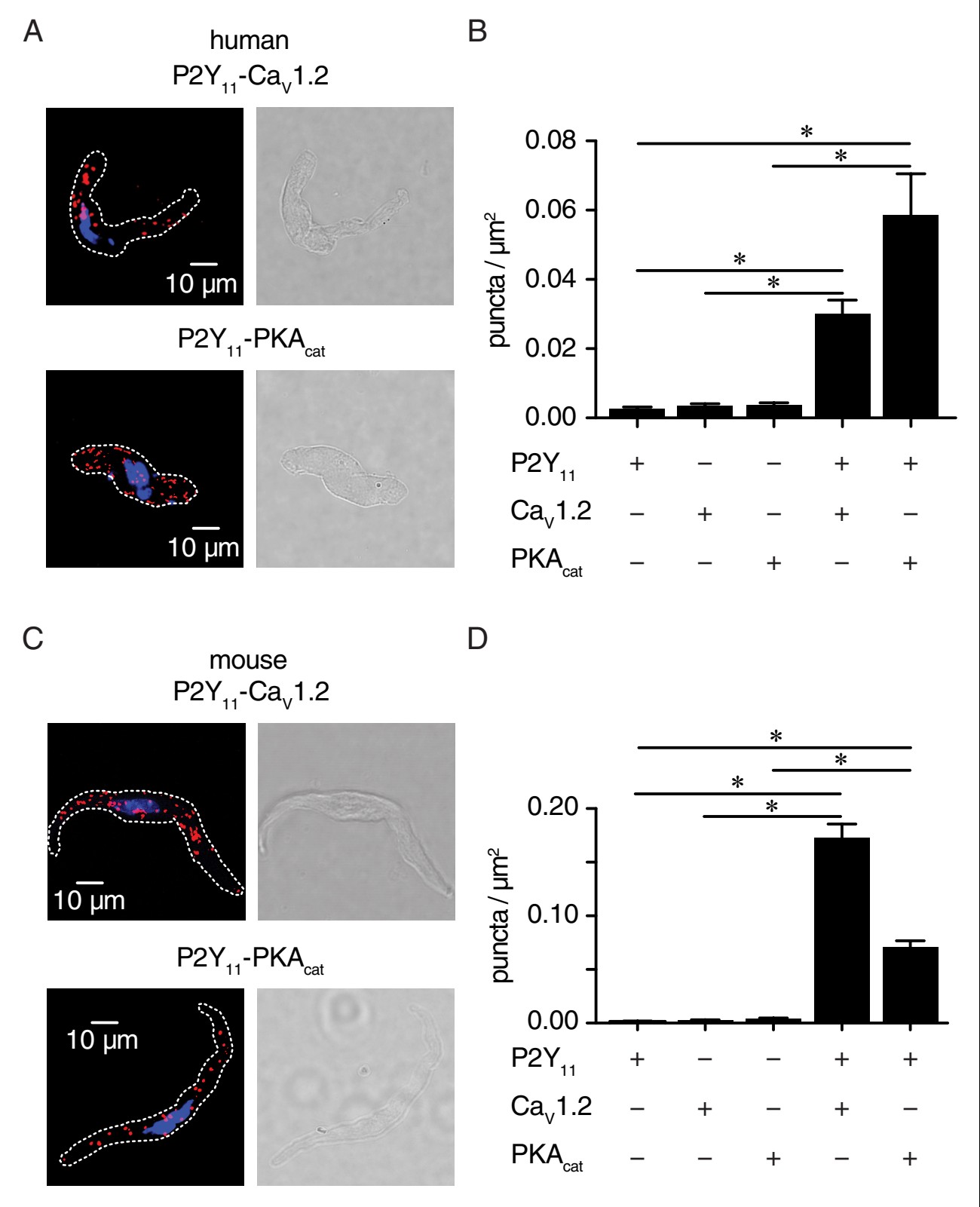

**Figure 4.** P2Y$_{11}$ associates with Ca$_V$1.2 and PKA$_{cat}$ in human and murine arterial myocytes. (**A**) Exemplary fluorescence PLA (red)/DAPI (blue) and differential interference contrast (right) images of human arterial myocytes labeled for P2Y$_{11}$ + Ca$_V$1.2 and P2Y$_{11}$ + PKA$_{cat}$. (**B**) Quantification of PLA fluorescent puncta per cell area (puncta/µm$^2$) for human arterial myocytes labeled for P2Y$_{11}$ (n = 26 cells from three human samples), Ca$_V$1.2 (n = 20 cells from three humans), PKA$_{cat}$ (n = 17 cells from three humans), P2Y$_{11}$ + Ca$_V$1.2 (n = 23 cells from three humans), and P2Y$_{11}$ + PKA$_{cat}$ (n = 20 cells

*Figure 4 continued on next page*

*Figure 4 continued*

from three humans) (*p<0.05, Kruskal-Wallis with Dunn's multiple comparisons; *Figure 4—source data 1*). (C) Representative fluorescence PLA (red)/ DAPI (blue) and differential interference contrast (right) images of mouse arterial myocytes labeled for $P2Y_{11}$ + $Ca_V1.2$ and $P2Y_{11}$ + $PKA_{cat}$. (D) Quantification of PLA fluorescent puncta per $\mu m^2$ cell area for mouse arterial myocytes labeled for $P2Y_{11}$ (n = 44 cells from six mice), $Ca_V1.2$ (n = 15 cells from six mice), $PKA_{cat}$ (n = 19 cells from six mice), $P2Y_{11}$ + $Ca_V1.2$ (n = 25 cells from six mice), and $P2Y_{11}$ + $PKA_{cat}$ (n = 29 cells from six mice) (*p<0.05, Kruskal-Wallis with Dunn's multiple comparisons; *Figure 4—source data 2*).
DOI: https://doi.org/10.7554/eLife.42214.030

The following source data and figure supplement are available for figure 4:

**Source data 1.** Excel spreadsheet containing the individual numeric values of puncta/$\mu m^2$ graphs in *Figure 4B*.
DOI: https://doi.org/10.7554/eLife.42214.032
**Source data 2.** Excel spreadsheet containing the individual numeric values of puncta/$\mu m^2$ graphs in *Figure 4D*.
DOI: https://doi.org/10.7554/eLife.42214.033
**Figure supplement 1.** Negative controls for PLA in human and murine arterial myocytes.
DOI: https://doi.org/10.7554/eLife.42214.031

*1*), robust fluorescent puncta were detected in human (*Figure 4A and B*) and mouse (*Figure 4C and D*) arterial myocytes co-labeled for $P2Y_{11}$ and $Ca_V1.2$ or $P2Y_{11}$ and $PKA_{cat}$. Altogether, these results suggest that human $P2Y_{11}$ or mouse $P2Y_{11}$-like receptors are located on the surface membrane of arterial myocytes within nanometer proximity ($\sim\leq40$ nm) of a subpopulation of $Ca_V1.2$ and $PKA_{cat}$.

## Increased sarcolemmal cAMP synthesis in arterial myocytes in response to a $P2Y_{11}$ agonist recapitulates glucose effects

$P2Y_{11}$ can stimulate adenylyl cyclase (AC) activity to produce cAMP (*Communi et al., 1997*; *Kennedy, 2017*; *Schuchardt et al., 2012*; *von Kügelgen and Harden, 2011*). Thus, we first used a membrane-targeted Epac1-camps-based FRET sensor (ICUE3-PM) (*Allen et al., 2006*; *Liu et al., 2011*) expressed in tsA-201 cells to validate the $P2Y_{11}$ pharmacology (*Figure 5—figure supplement 1A*) (*Dreisig and Kornum, 2016*). As expected, the highly selective $P2Y_{11}$ agonist NF546 (500 nM) (*Meis et al., 2010*) increased cAMP synthesis in these cells. This NF546-induced cAMP response remained intact in cells pretreated with either selective $P2Y_1$ (10 $\mu M$ MRS2179) or $P2Y_6$ (100 nM MRS2578) inhibitors but was prevented in cells exposed to the selective $P2Y_{11}$ inhibitor NF340 (10 $\mu M$; 500-fold more selective for $P2Y_{11}$ than other P2Y receptors) (*Meis et al., 2010*).

We recapitulated the NF546-induced cAMP response and its ablation by pretreatment with NF340 in primary, unpassaged human arterial myocytes expressing the ICUE3-PM sensor (*Figure 5A and B*). Increasing D-glucose concentration from 5 mM to 15 mM, which are glucose concentrations comparable to those observed in nondiabetic and diabetic patients respectively, induced a small yet significant increase in cAMP synthesis in human cells (*Figure 5A and B*). The glucose effects are not due to osmotic changes as equimolar concentrations of the non-permeable mannitol did not induce any cAMP synthesis (*Figure 5—figure supplement 1B*). As expected, the synthesis of cAMP was further amplified by the broad AC activator forskolin (1 $\mu M$) (*Figure 5A*) even in the presence of 20 mM mannitol (*Figure 5—figure supplement 1B*). Simultaneous application of 15 mM D-glucose +NF546 resulted in cAMP synthesis of a similar magnitude to that induced by independent treatments (*Figure 5A and B*). This suggests that the effects of elevated glucose and the NF546 compound may be acting through a similar pathway that requires activation of $P2Y_{11}$. Consistent with this notion, application of NF546 by itself or 15 mM D-glucose +NF546 failed to induce cAMP synthesis in cells pretreated with NF340 (*Figure 5A and B*). Yet forskolin, which bypasses $P2Y_{11}$, was still able to stimulate global cAMP production. Similar results were observed in murine arterial myocytes expressing the ICUE3-PM sensor (*Figure 5C and D* and *Figure 5—figure supplement 1C*). Intriguingly, cAMP synthesis in response to elevated glucose and the $P2Y_{11}$ agonist NF546 was significantly larger in human myocytes compared to mouse cells (glucose: $0.077 \pm 0.003$ in human vs $0.045 \pm 0.002$ in mouse; NF546: $0.083 \pm 0.005$ in human vs $0.045 \pm 0.004$ in mouse; p<0.05). The reasons for this are currently unclear but may include intrinsic differences associated with species (human vs mouse), blood vessels (human adipose arteries vs mouse aortae) and expression patterns of the biosensor in human vs mouse cells (*Reddy et al., 2018*), as well as variations in protein expression and receptor/enzyme activity (e.g. $P2Y_{11}$, AC and PDE expression/activity) in human vs mouse arterial myocytes. Whether either or all of these factors contribute to distinctive

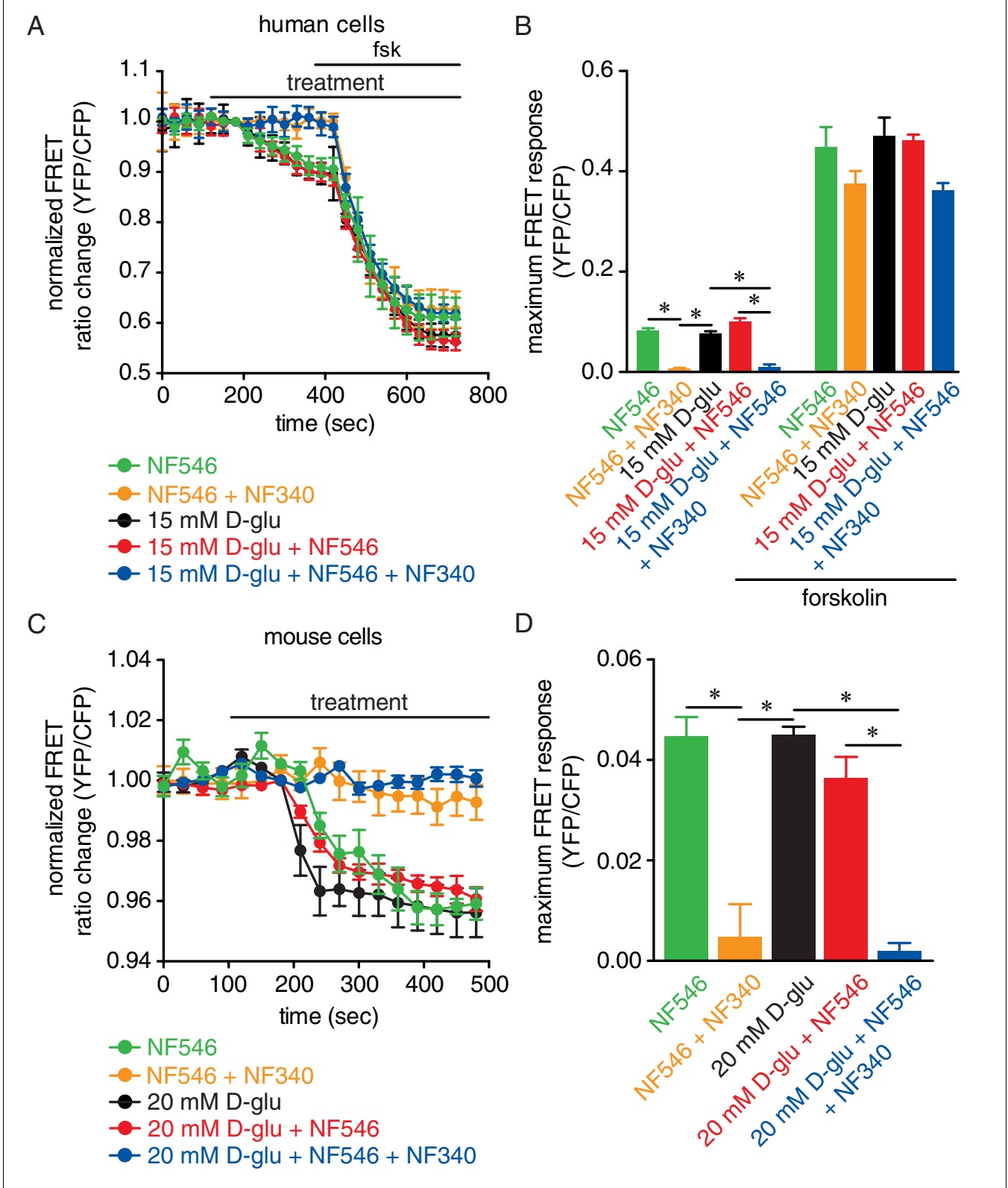

**Figure 5.** Glucose and the P2Y$_{11}$ agonist NF546 increase sarcolemmal cAMP synthesis in arterial myocytes, and these effects are prevented by the P2Y$_{11}$ antagonist NF340. (**A**) Time course of YFP/CFP (donor/acceptor) FRET ratios (normalized to resting levels before treatment) in human arterial myocytes expressing the ICUE3-PM biosensor in response to 15 mM D-glucose (black; n = 19 cells from two humans), 500 nM NF546 (green; n = 18 cells from two humans) and 15 mM D-glucose +500 nM NF546 (red; n = 14 cells from two humans) before and after application of broad adenylyl

*Figure 5 continued on next page*

*Figure 5 continued*

cyclase agonist forskolin (1 µM). In a set of experiments, cells were first pre-treated with the P2Y$_{11}$ antagonist NF340 (10 µM) for at least 15–20 min before treatment with 500 nM NF546 (orange; n = 15 cells from two humans) and 15 mM D-glucose +500 nM NF546 (blue; n = 13 cells from two humans). Horizontal bars indicate treatment. Increases in cAMP production are represented by decreases in YFP/CFP ratio due to binding of cAMP to the biosensor. (B) Bar plot of maximum FRET responses (YFP/CFP) for human arterial myocytes in response to the indicated treatment (*p<0.05, Kruskal-Wallis with Dunn's multiple comparisons; *Figure 5—source data 1*. Statistical differences were compared between 15 mM D-glucose vs NF546, 15 mM D-glucose vs 15 mM D-glucose +NF546, 15 mM D-glucose vs NF546 +NF340, 15 mM D-glucose vs 15 mM D-glucose +NF546+NF340, NF546 vs NF546 +NF340, NF546 vs 15 mM D-glucose +NF546, NF546 vs 15 mM D-glucose +NF546+NF340, 15 mM D-glucose +NF546 vs 15 mM D-glucose +NF546+NF340). (C) Time course of YFP/CFP (donor/acceptor) FRET ratios (normalized to resting levels before treatment) in mouse arterial myocytes expressing the ICUE3-PM biosensor in response to 20 mM D-glucose (black; n = 83 cells from three mice), 500 nM NF546 (green; n = 13 cells from three mice) and 20 mM D-glucose +500 nM NF546 (red; n = 13 cells from three mice). Horizontal bars indicate treatment. In a set of experiments, cells were first pre-treated with the P2Y$_{11}$ antagonist NF340 (10 µM) for at least 15–20 min before treatment with 500 nM NF546 (orange; n = 17 cells from three mice) and 20 mM D-glucose +500 nM NF546 (blue; n = 18 cells from three mice). Increases in cAMP production are represented by decreases in YFP/CFP ratio due to binding of cAMP to the biosensor. (D) Bar plot of maximum FRET responses (YFP/CFP) for mouse arterial myocytes in response to the indicated treatment (*p<0.05, Kruskal-Wallis with Dunn's multiple comparisons; *Figure 5—source data 2*. Statistical differences were compared between all datasets).

DOI: https://doi.org/10.7554/eLife.42214.034

The following source data and figure supplements are available for figure 5:

**Source data 1.** Excel spreadsheet containing the individual numeric values for maximum FRET responses in *Figure 5B*.
DOI: https://doi.org/10.7554/eLife.42214.039

**Source data 2.** Excel spreadsheet containing the individual numeric values for maximum FRET responses in *Figure 5D*.
DOI: https://doi.org/10.7554/eLife.42214.040

**Figure supplement 1.** The P2Y$_{11}$ agonist NF546 increases cAMP and this is blocked in the presence of the P2Y$_{11}$ inhibitor NF340 but not with the P2Y$_{1}$ inhibitor MSR2179 or P2Y$_{6}$ inhibitor MRS2578 in tsA-201 cells, and no increases in cAMP in arterial myocytes with mannitol.
DOI: https://doi.org/10.7554/eLife.42214.035

**Figure supplement 1—source data 1.** Excel spreadsheet containing the individual numeric values for maximum FRET responses in *Figure 5—figure supplement 1A*.
DOI: https://doi.org/10.7554/eLife.42214.036

**Figure supplement 1—source data 2.** Excel spreadsheet containing the individual numeric values for maximum FRET responses in *Figure 5—figure supplement 1B*.
DOI: https://doi.org/10.7554/eLife.42214.037

**Figure supplement 1—source data 3.** Excel spreadsheet containing the individual numeric values for maximum FRET responses in *Figure 5—figure supplement 1C*.
DOI: https://doi.org/10.7554/eLife.42214.038

cAMP responses in human vs mouse arterial myocytes as well as their physiological implications remain to be determined. Nevertheless, our findings suggest a role for human P2Y$_{11}$ and mouse P2Y$_{11}$-like receptors in glucose-induced cAMP synthesis.

## The P2Y$_{11}$ inhibitor NF340 prevents glucose-induced Ser[1928] phosphorylation, LTCC activity and vasoconstriction

In our next series of experiments, we confirmed the glucose-induced increase in I$_{Ba}$ detected in mouse arterial myocytes (*Figure 1B*) using freshly dissociated human arterial myocytes in which external glucose was elevated from 5 mM to 15 mM (*Figure 6A and B*). These results indicate that elevations in extracellular glucose also stimulate LTCC activity in human arterial myocytes (*Nystoriak et al., 2017*). The increase in I$_{Ba}$ evoked by 15 mM D-glucose was prevented in human cells pretreated with NF340 (*Figure 6A and B*). Glucose-induced potentiation of I$_{Ba}$ was also prevented in mouse arterial myocytes pretreated with NF340 (*Figure 6C and D*). However, inhibition of P2Y$_{6}$ receptors with MRS2578, which has been implicated in myogenic tone regulation and glucose-induced NFAT activation in arterial myocytes (*Brayden et al., 2013*; *Nilsson et al., 2006*), failed to stop the increase in I$_{Ba}$ in response to 20 mM D-glucose (*Figure 6C and D*). Note that basal I$_{Ba}$ were similar in cells exposed to 10 mM D-glucose and 10 mM D-glucose +MRS2578, suggesting that inhibition of P2Y$_{6}$ did not alter basal LTCC activity (*Figure 6C and D*). Altogether, these results suggest that P2Y$_{6}$ receptors are not involved in glucose-induced potentiation of LTCC activity, and that these glucose effects are prevented with a P2Y$_{11}$ inhibitor.

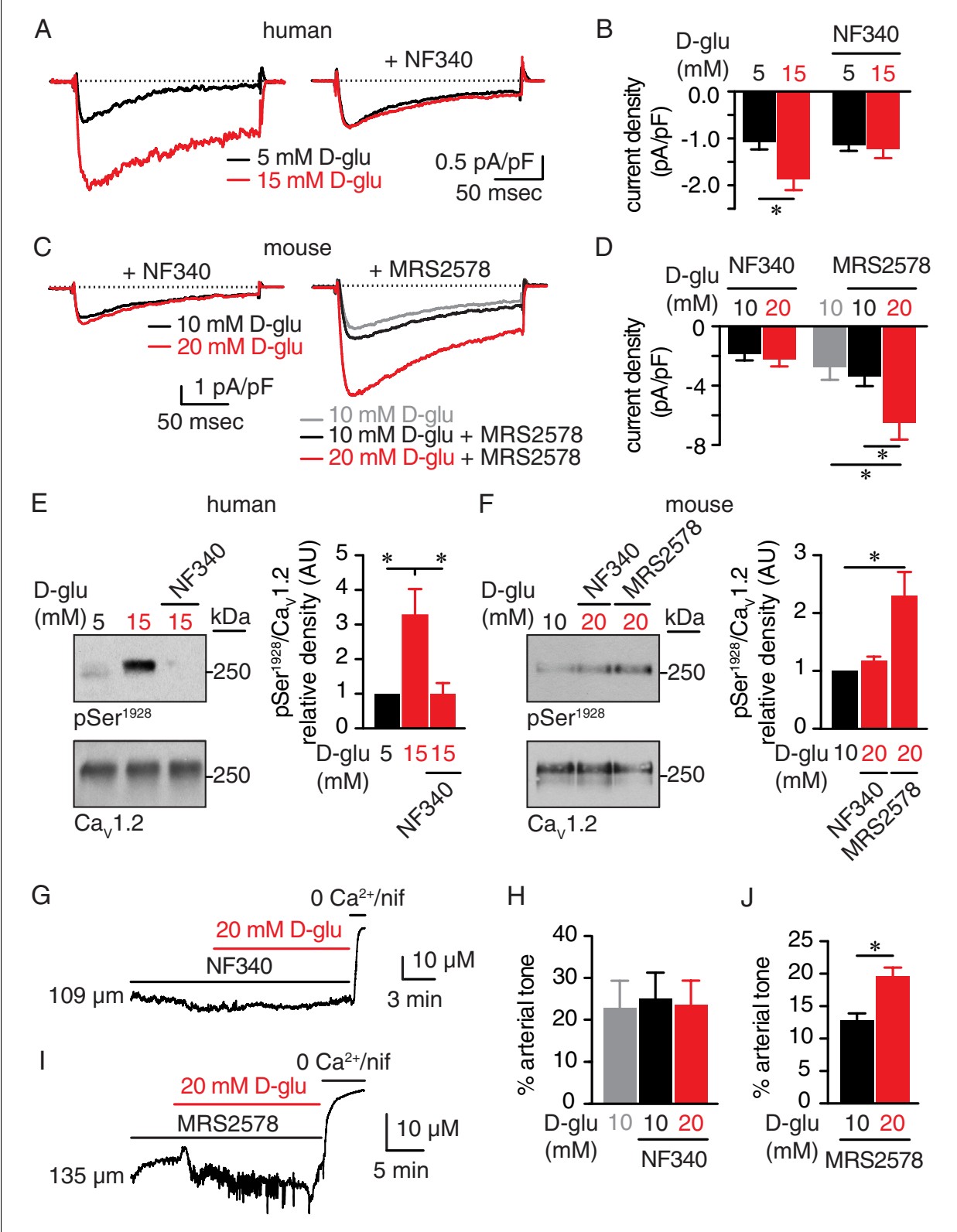

**Figure 6.** The P2Y[11] inhibitor NF340 prevents glucose-induced elevations in Ser[1928] phosphorylation, LTCC activity, and vasoconstriction in human and murine arterial myocytes. (**A**) Representative $I_{Ba}$ recordings from the same cell and (**B**) summary current density data obtained from freshly dissociated human arterial myocytes before and after increasing extracellular D-glucose from 5 mM to 15 mM in the absence (n = 12 cells from six humans) and presence (n = 10 cells from five humans) of NF340 (10 µM) (*p<0.05, paired *t* test; *Figure 6—source data 1*). (**C**) Exemplary $I_{Ba}$ traces from the same cell

*Figure 6 continued on next page*

*Figure 6 continued*

and (D) amalgamated current density data recorded from mouse cerebral arterial myocytes treated with 10 µM NF340 (n = 8 cells from three mice) or 100 nM MRS2578 (n = 7 cells from five mice) before and after increasing extracellular D-glucose from 10 mM to 20 mM (*p<0.05, Kruskal-Wallis with Dunn's multiple comparisons; *Figure 6—source data 2*). Representative immunoreactive bands for phosphorylated Ser[1928] (pSer[1928]) and total $Ca_V1.2$ and densitometry for pSer[1928]/$Ca_V1.2$ ratio in lysates from (E) human arteries exposed to 5 mM and 15 mM D-glucose in the absence and presence of 10 µM NF340 (n = 6 arterial lysates per condition; *p<0.05, Kruskal-Wallis with Dunn's multiple comparisons; *Figure 6—source data 3*) and (F) mouse cerebral and mesenteric arteries exposed to 10 mM or 20 mM D-glucose in the presence of 10 µM NF340 or 100 nM MRS2578 (n = 4 arterial lysates per condition; *p<0.05 Kruskal-Wallis with Dunn's multiple comparisons; *Figure 6—source data 4*). Representative diameter recordings and summary data from pressurized (60 mmHg) mouse cerebral arteries exposed to (G–H) 10 µM NF340 (n = 4 arteries from four mice; Friedman with Dunn's multiple comparisons; *Figure 6—source data 5*) or (I–J) 100 nM MRS2578 (n = 7 arteries from seven mice; *p<0.05, Wilcoxon matched pair test; *Figure 6—source data 6*) before and after application of 20 mM D-glucose.

DOI: https://doi.org/10.7554/eLife.42214.041

The following source data and figure supplements are available for figure 6:

**Source data 1.** Excel spreadsheet containing the individual numeric values of current density analyzed in *Figure 6B*.
DOI: https://doi.org/10.7554/eLife.42214.046

**Source data 2.** Excel spreadsheet containing the individual numeric values of current density analyzed in *Figure 6D*.
DOI: https://doi.org/10.7554/eLife.42214.047

**Source data 3.** Excel spreadsheet containing the individual numeric values of pSer[1928]/$Ca_V1.2$ relative density analyzed in *Figure 6E*.
DOI: https://doi.org/10.7554/eLife.42214.048

**Source data 4.** Excel spreadsheet containing the individual numeric values of pSer[1928]/$Ca_V1.2$ relative density analyzed in *Figure 6F*.
DOI: https://doi.org/10.7554/eLife.42214.049

**Source data 5.** Excel spreadsheet containing the individual numeric values of % arterial tone analyzed in *Figure 6H* and corresponding raw diameters.
DOI: https://doi.org/10.7554/eLife.42214.050

**Source data 6.** Excel spreadsheet containing the individual numeric values of % arterial tone analyzed in *Figure 6J* and corresponding raw diameters.
DOI: https://doi.org/10.7554/eLife.42214.051

**Figure supplement 1.** Full-length blots for *Figure 6E and F*, high K[+] induced constriction in arteries pretreated with NF340 and MRS2578 corresponding to data in *Figure 6*, and glucose-induced vasoconstriction is not prevented or inhibited by a selective P2Y$_1$ antagonist.
DOI: https://doi.org/10.7554/eLife.42214.042

**Figure supplement 1—source data 1.** Excel spreadsheet containing the individual numeric values of 60 mM K[+]-induced % constriction analyzed in *Figure 6—figure supplement 1C*.
DOI: https://doi.org/10.7554/eLife.42214.043

**Figure supplement 1—source data 2.** Excel spreadsheet containing the individual numeric values of % arterial tone analyzed in *Figure 6—figure supplement 1D* and corresponding raw diameters.
DOI: https://doi.org/10.7554/eLife.42214.044

**Figure supplement 1—source data 3.** Excel spreadsheet containing the individual numeric values of % arterial tone analyzed in *Figure 6—figure supplement 1E* and corresponding raw diameters.
DOI: https://doi.org/10.7554/eLife.42214.045

Since the P2Y$_{11}$ inhibitor hindered the glucose effects on LTCC activity, we also explored whether the NF340 compound interfered with the Ser[1928] phosphorylation state of the channel and vascular reactivity in response to hyperglycemia. As observed in mouse arteries (*Figure 1C*), we found that elevating extracellular glucose significantly increased Ser[1928] phosphorylation in human arterial lysates (*Figure 6E*; *Figure 6—figure supplement 1A*). Yet, this glucose-mediated increase in Ser[1928] phosphorylation was prevented if arterial lysates were pretreated with NF340, whereas phosphorylation still occurred if arterial lysates were pretreated with the P2Y$_6$ inhibitor MRS2578 (*Figure 6E and F*, and *Figure 6—figure supplement 1A and B*). Furthermore, vasoconstriction in response to elevated glucose was blocked in pressurized (60 mmHg) mouse cerebral arteries exposed to NF340 (*Figure 6G and H* and *Supplementary file 2*) but not to MRS2578 (*Figure 6I and J* and *Supplementary file 2*). Note that while arterial tone was similar in control and NF340-treated arteries, it was reduced in arteries exposed to MRS2578 (*Figure 6I and J*), which is in line with recent data showing that P2Y$_6$ is a sensor of pressure-induced constriction (*Brayden et al., 2013*). Vasoconstriction in response to 60 mM K[+] was similar in control and NF340 as well as MRS2578-treated arteries (*Figure 6—figure supplement 1C* and *Supplementary file 1* and *2*). We also evaluated a potential role for P2Y$_1$ receptors in stimulating PKA-mediated vasoconstriction as these receptors have been suggested to crosstalk with P2Y$_{11}$ to regulate their function in HEK cells and neurons (*del Puerto et al., 2012*; *King and Townsend-Nicholson, 2008*). Data revealed that the glucose-

induced vasoconstriction was not altered by adding the selective P2Y$_1$ inhibitor MRS2179 (10 µM) either before or after 20 mM D-glucose stimulation (*Figure 6—figure supplement 1D and E* and *Supplementary file 1* and *2*). These results suggest that the P2Y$_1$ receptor is not involved in glucose-mediated vasoconstriction. Together, these data demonstrate the effectiveness of a P2Y$_{11}$ inhibitor in blocking the remodeling of Ser$^{1928}$ phosphorylation and LTCC activity that mediates vasoconstriction in response to elevated glucose.

## The P2Y$_{11}$ agonist NF546 stimulates Ser$^{1928}$ phosphorylation, PKA-dependent LTCC activity and vasoconstriction

We next investigated whether a selective P2Y$_{11}$ agonist could recapitulate the glucose-mediated Ser$^{1928}$ phosphorylation and PKA-dependent LTCC activity. Using freshly dissociated human arterial myocytes, we found that I$_{Ba}$ was significantly increased by application of NF546 (*Figure 7A*). This P2Y$_{11}$ agonist also increased I$_{Ba}$ at multiple membrane potentials in mouse arterial myocytes with no change in the current-voltage (I-V) relationship (V$_{max}$ = 14.9 ± 2.0 mV for 10 mM D-glucose and V$_{max}$ = 10.9 ± 1.1 mV for NF546; p=0.1085, extra sum-of-squares *F* test) (*Figure 7B*). The NF546-mediated increase in I$_{Ba}$ was similar in magnitude to that induced by elevated glucose (*Figure 7—figure supplement 1A*). Similar I$_{Ba}$ potentiation was also observed after application of the non-hydrolyzed ATP analog and potent P2Y$_{11}$ agonist ATPγS (1 µM; *Figure 7—figure supplement 1B*) (*von Kügelgen and Harden, 2011*). Note that ATPγS stimulates P2Y$_{11}$ activity with EC$_{50}$ that ranges from 31 nM to 23 µM, depending on readout, cell type and species used, but it is still a more potent activator than ATP itself (*Communi et al., 1999*; *Haas et al., 2014*; *Kennedy, 2017*; *Meis et al., 2010*; *Qi et al., 2001*; *Schuchardt et al., 2012*; *von Kügelgen and Harden, 2011*). We further correlated the potentiation in I$_{Ba}$ by NF546 to an elevation of Ser$^{1928}$ phosphorylation in freshly dissected human and mouse arterial lysates (*Figure 7C* and *Figure 7—figure supplement 1C*). Since PKA is essential for glucose-mediated remodeling of I$_{Ba}$ in arterial myocytes and can be activated downstream of P2Y$_{11}$ (*Navedo et al., 2010b*; *Nystoriak et al., 2017*), we also examined its involvement in NF546-mediated potentiation of LTCC activity in arterial myocytes. Consistent with a key role for PKA, NF546 failed to upregulate I$_{Ba}$ in arterial myocytes incubated with the PKA inhibitors PKI (100 nM) or rpcAMP (10 µM) (*Figure 7D*).

Having established that application of the P2Y$_{11}$ agonist NF546 influences Ser$^{1928}$ phosphorylation state as well as LTCC activity, we investigated whether it could also modulate arterial tone. Accordingly, pressurized (60 mmHg) arteries that developed stable arterial tone and responded to 60 mM K$^+$ (*Figure 7—figure supplement 1D* and *Supplementary file 1* and *2*) showed significant constriction upon NF546 application (*Figure 7E*). Interestingly, NF546 did not induce further constriction in wt arteries previously exposed to 20 mM D-glucose (*Figure 7F*). Indeed, arterial tone was similar in arteries treated with 20 mM D-glucose or 20 mM D-glucose +NF546 (*Supplementary file 1* and *2*). Confirming a critical role for Ser$^{1928}$, arteries from a S1928A knockin mouse in which phosphorylation of this Ca$_V$1.2 amino acid is prevented (*Lemke et al., 2008*; *Nystoriak et al., 2017*; *Qian et al., 2017*), failed to constrict to NF546 (*Figure 7F* and *Supplementary file 1* and *2*). Response of S1928A arteries to 60 mM K$^+$ was similar to those observed for wt arteries (*Figure 7—figure supplement 1D* and *Supplementary file 1* and *2*). These results suggest that a P2Y$_{11}$ agonist can induce Ser$^{1928}$ phosphorylation, PKA-dependent LTCC potentiation and vasoconstriction, thus recapitulating glucose effects (*Navedo et al., 2010b*; *Nystoriak et al., 2017*), and providing further evidence of the involvement of human P2Y$_{11}$ and mouse P2Y$_{11}$-like receptors in these alterations.

## Increased Ser$^{1928}$ phosphorylation and LTCC activity during chronic diabetic hyperglycemia are prevented by a P2Y$_{11}$ inhibitor

To explore whether a P2Y$_{11}$ inhibitor could prevent changes in Ser$^{1928}$ phosphorylation and LTCC activity during chronic diabetic hyperglycemia, we exposed isolated arteries for 48 hr to 10 mM D-glucose, 20 mM mannitol, 20 mM D-glucose or 20 mM D-glucose +NF340. This organ culture method prevents arterial myocyte phenotypic changes associated with prolonged culturing conditions. Furthermore, the exposure time is sufficient to induce many of the vascular remodeling phenomena associated with elevated glucose in arterial myocytes, including increased LTCC activity, decreased K$^+$ channel activity and downstream activation of Ca$^{2+}$-dependent transcription factors

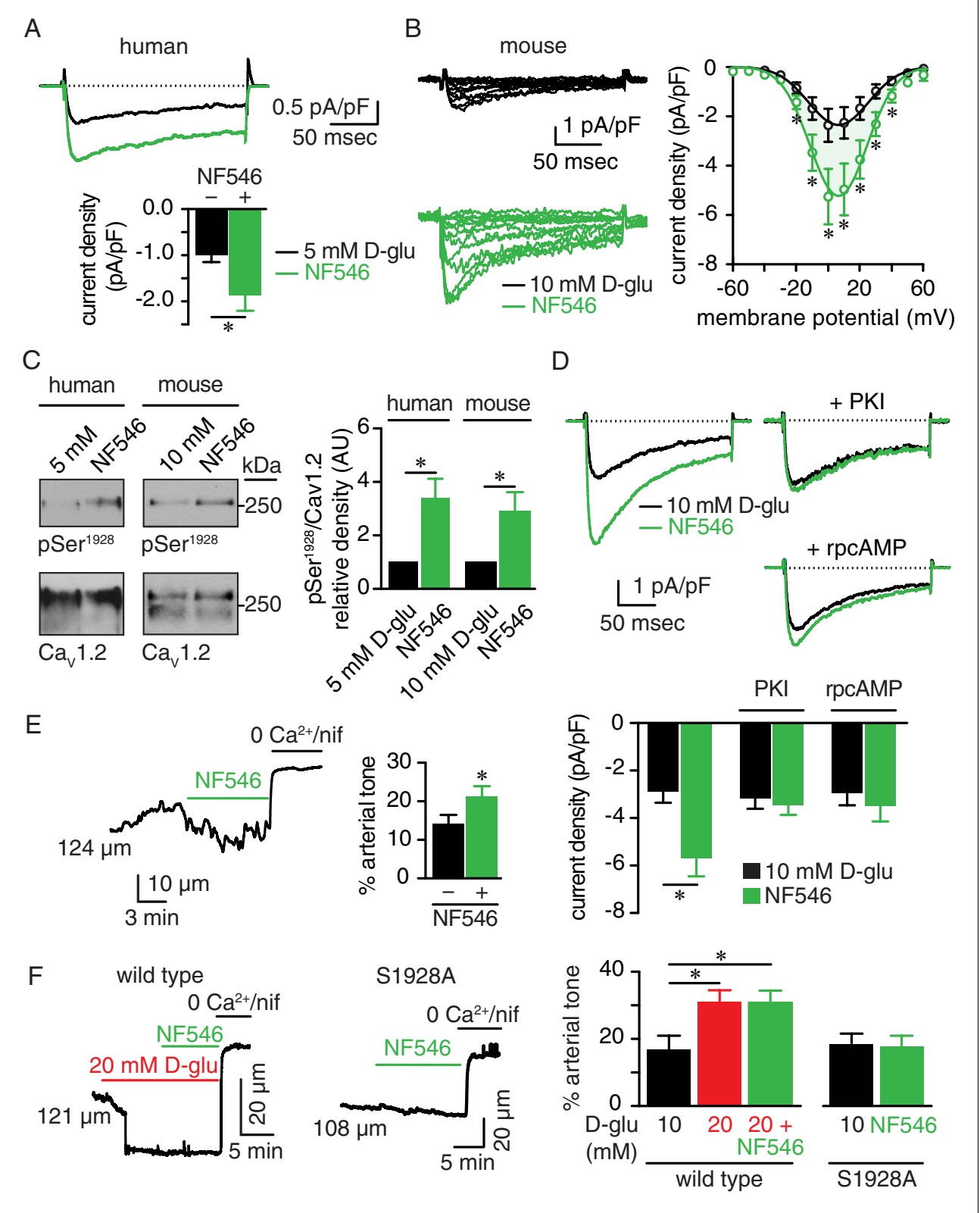

**Figure 7.** The P2Y[11] agonist NF546 increases Ser[1928] phosphorylation, LTCC activity, and induces vasoconstriction. (**A**) Representative I[Ba] recordings from the same cell (*top*) and summary I[Ba] data (*bottom*) from freshly dissociated human arterial myocytes in response to step depolarizations from −70 to +10 mV before and after application of 500 nM NF546 (n = 11 cells from five humans; *p<0.05, paired *t* test; ***Figure 7—source data 1***). (**B**) Representative I[Ba] recordings from the same cell (*left*) triggered by step depolarization from −70 mV to voltages ranging from −60 to +60 mV before

*Figure 7 continued on next page*

*Figure 7 continued*

and after application of 500 nM NF546 in mouse cerebral arterial myocytes and corresponding $I_{Ba}$-voltage relationship (*right*) (n = 8 cells from five mice) (*p<0.05, paired *t* test; ***Figure 7—source data 2***). (C) Exemplary immunoblot detection of phosphorylated Ser[1928] (pSer[1928]) and total $Ca_V1.2$ from human (*left*) and mouse cerebral and mesenteric arteries (right) incubated with 500 nM NF546 and respective densitometry quantification of pSer[1928]/ $Ca_V1.2$ ratio (n = 6 arterial lysates per condition for humans; n = 6 arterial lysates per condition for mice) (*p<0.05, Wilcoxon matched pairs test; ***Figure 7—source data 3***). (D) Representative $I_{Ba}$ recordings from the same cell (top) and summary $I_{Ba}$ data (bottom) from mouse arterial myocytes evoked by step depolarizations from −70 to +10 mV before and after application of 500 nM NF546 in the absence (n = 9 cells, four mice) and presence of 100 nM PKI (n = 9 cells, five mice) or 10 μM rpcAMP (n = 9 cells, four mice) (*p<0.05, paired *t* test; ***Figure 7—source data 4***). (E) Representative diameter recording and summary arterial tone data from pressurized (60 mmHg) mouse cerebral arteries exposed to 500 nM NF546 (n = 6 arteries, six mice) (*p<0.05, Wilcoxon matched pairs test; ***Figure 7—source data 5***). (F) Representative diameter recordings and summary arterial tone data from pressurized (60 mmHg) wt mouse cerebral arteries exposed to 20 mM D-glucose before and after application of 500 nM NF546 (n = 6 arteries, six mice, *left*; *p<0.05, Friedman with Dunn's multiple comparisons; ***Figure 7—source data 6***) and S9128A mouse cerebral arteries after NF546 application (n = 4 from three mice, right; ***Figure 7—source data 6***).

DOI: https://doi.org/10.7554/eLife.42214.052

The following source data and figure supplements are available for figure 7:

**Source data 1.** Excel spreadsheet containing the individual numeric values of current density analyzed in ***Figure 7A***.

DOI: https://doi.org/10.7554/eLife.42214.057

**Source data 2.** Excel spreadsheet containing the individual numeric values of current density analyzed in ***Figure 7B***.

DOI: https://doi.org/10.7554/eLife.42214.058

**Source data 3.** Excel spreadsheet containing the individual numeric values of pSer[1928]/$Ca_V1.2$ relative density analyzed in ***Figure 7C***.

DOI: https://doi.org/10.7554/eLife.42214.059

**Source data 4.** Excel spreadsheet containing the individual numeric values of current density analyzed in ***Figure 7D***.

DOI: https://doi.org/10.7554/eLife.42214.060

**Source data 5.** Excel spreadsheet containing the individual numeric values of % arterial tone analyzed in ***Figure 7E*** and corresponding raw diameters.

DOI: https://doi.org/10.7554/eLife.42214.061

**Source data 6.** Excel spreadsheet containing the individual numeric values of % arterial tone analyzed in ***Figure 7F*** and corresponding raw diameters.

DOI: https://doi.org/10.7554/eLife.42214.062

**Figure supplement 1.** D-glucose and NF546 elicit changes in $I_{Ba}$ of similar magnitude in mouse arterial myocytes, ATPγS increases $I_{Ba}$ in arterial myocytes, full-length blots corresponding to data in ***Figure 7C***, and high $K^+$-induced constriction in wt and S1928A arteries treated with NF546 corresponding to data in ***Figure 7***.

DOI: https://doi.org/10.7554/eLife.42214.053

**Figure supplement 1—source data 1.** Excel spreadsheet containing the individual numeric values of change in current density analyzed in ***Figure 7— figure supplement 1A***.

DOI: https://doi.org/10.7554/eLife.42214.054

**Figure supplement 1—source data 2.** Excel spreadsheet containing the individual numeric values of current density analyzed in ***Figure 7—figure supplement 1B***.

DOI: https://doi.org/10.7554/eLife.42214.055

**Figure supplement 1—source data 3.** Excel spreadsheet containing the individual numeric values of 60 mM $K^+$-induced % constriction analyzed in ***Figure 7—figure supplement 1D***.

DOI: https://doi.org/10.7554/eLife.42214.056

(***Navedo et al., 2010b***; ***Nieves-Cintrón et al., 2015***; ***Nilsson et al., 2006***; ***Nystoriak et al., 2014***; ***Nystoriak et al., 2017***). LTCC function was assessed by measuring single-channel activity using the cell-attached configuration from dissociated arterial myocytes under each experimental condition described above. We found that wt cells isolated from the 20 mM D-glucose but not the 20 mM mannitol treated group had significantly enhanced LTCC open probability ($nP_o$) compared to those in the 10 mM D-glucose group (***Figure 8A and B***). This increase in LTCC activity during diabetic hyperglycemia, however, was prevented in arterial myocytes from arteries treated in the presence of the $P2Y_{11}$ inhibitor NF340. Ser[1928] phosphorylation was significantly elevated in arteries chronically exposed to 20 mM D-glucose, but this effect was completely abolished in the presence of NF340 (***Figure 8C*** and ***Figure 8—figure supplement 1A***). Consistent with a key role for Ser[1928] phosphorylation in potentiation of LTCC activity in response to chronic elevations in glucose, LTCC $nP_o$ was similar in arterial myocytes from S1928A arteries incubated for 48 hr in 10 mM or 20 mM D-glucose (***Figure 8—figure supplement 1B***). These results suggest that treatment with a $P2Y_{11}$ inhibitor can

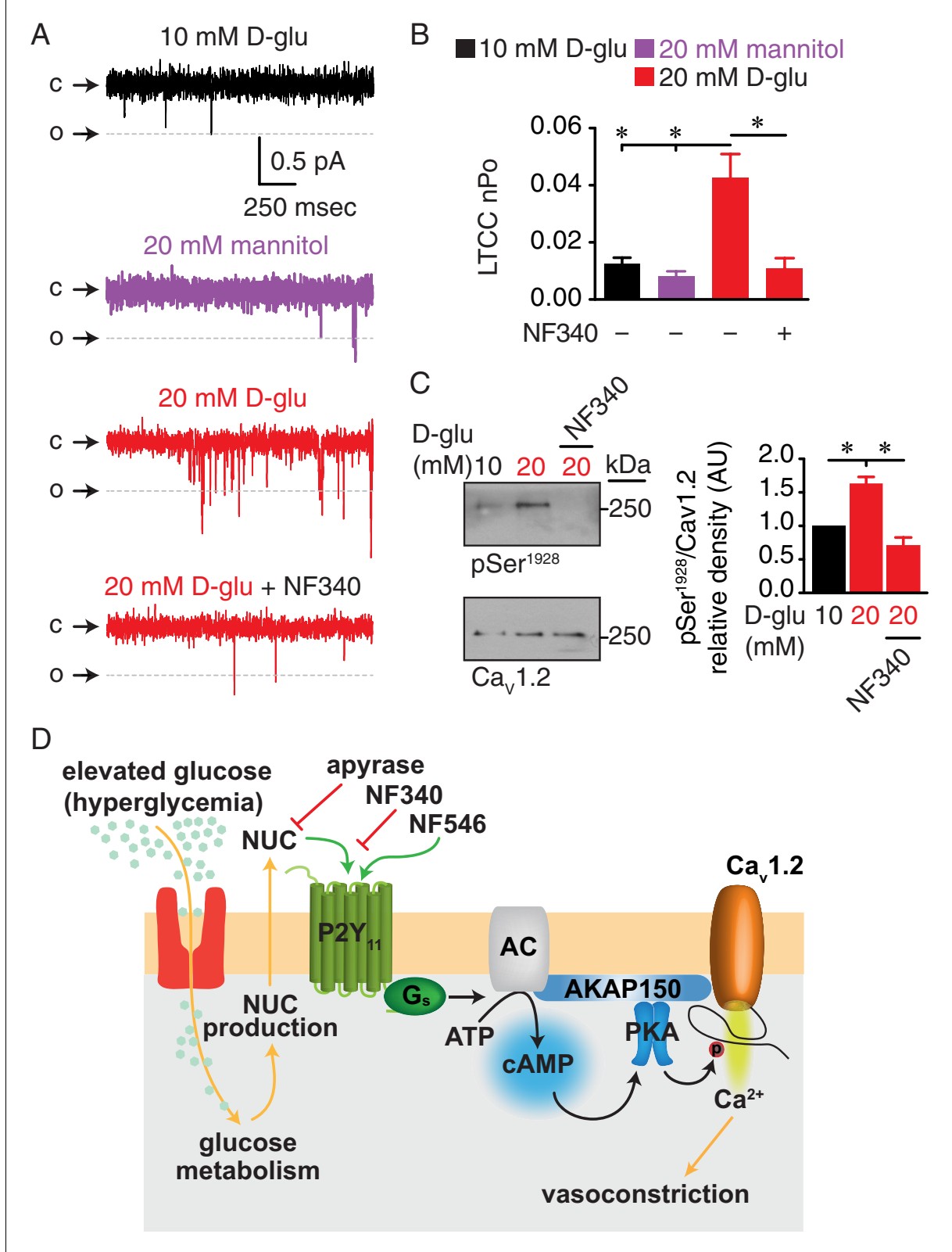

**Figure 8.** Augmented LTCC activity and Ser[1928] phosphorylation in response to chronic extracellular glucose elevations are prevented in the presence of the P2Y[11] antagonist NF340. (**A**) Representative single LTCC recordings obtained during a 2 s step depolarization from −80 to −30 mV and (**B**) bar plot of LTCC $nP_o$ in arterial myocytes isolated from mouse cerebral arteries incubated for 48 hr in 10 mM D-glucose (n = 10 cells from three mice), 20 mM mannitol (n = 13 cells from four mice), 20 mM D-glucose (n = 10 cells from four mice) and 20 mM D-glucose +10 μM NF340 (n = 10 cells from four

*Figure 8 continued on next page*

*Figure 8 continued*

mice). Channel openings (o) are represented by downward deflections from baseline (c) (*p<0.05, one-way ANOVA with Tukey post hoc test; *Figure 8—source data 1*). (C) Representative immunoblot detection of phosphorylated $Ser^{1928}$ ($pSer^{1928}$) and total $Ca_V1.2$ from mouse cerebral and mesenteric arteries incubated for 48 hr in 10 mM D-glucose, 20 mM D-glucose and 20 mM D-glucose +10 μM NF340 and densitometry quantification of $pSer^{1928}/Ca_V1.2$ ratio (n = 7 arterial lysates per condition) (*p<0.05, Kruskal-Wallis with Dunn's multiple comparisons; *Figure 8—source data 2*). (D) Proposed model for the role of $P2Y_{11}$ in PKA-dependent stimulation of LTCC activity and vasoconstriction during diabetic hyperglycemia (NUC = nucleotides).

DOI: https://doi.org/10.7554/eLife.42214.063

The following source data and figure supplements are available for figure 8:

**Source data 1.** Excel spreadsheet containing the individual numeric values of LTCC $nP_o$ analyzed in *Figure 8B*.
DOI: https://doi.org/10.7554/eLife.42214.066

**Source data 2.** Excel spreadsheet containing the individual numeric values of $pSer^{1928}/Ca_V1.2$ relative density analyzed in *Figure 8C*.
DOI: https://doi.org/10.7554/eLife.42214.067

**Figure supplement 1.** Full-length blots corresponding to data in *Figure 8C* and unchanged LTCC $nP_o$ in response to chronic elevations in extracellular glucose in S1928A arterial myocytes.
DOI: https://doi.org/10.7554/eLife.42214.064

**Figure supplement 1—source data 1.** Excel spreadsheet containing the individual numeric values of LTCC $nP_o$ analyzed in *Figure 8—figure supplement 1B*.
DOI: https://doi.org/10.7554/eLife.42214.065

avert PKA-mediated $Ser^{1928}$ phosphorylation and stimulated LTCC activity during diabetic hyperglycemia.

## Discussion

Increased arterial myocyte contractility and vasoconstriction in response to elevated extracellular glucose contribute to vascular complications in diabetes (*Cooper et al., 2001*; *Fleischhacker et al., 1999*; *Jackson et al., 2016*; *Montero et al., 2013*; *Nieves-Cintrón et al., 2015*; *Nystoriak et al., 2017*). This endothelium-independent component is attributed, at least in part, to glucose-induced decreases in $K^+$ channel expression and activity (*Jackson et al., 2016*; *Nieves-Cintrón et al., 2015*; *Nieves-Cintrón et al., 2017*; *Nieves-Cintrón et al., 2018*; *Nystoriak et al., 2014*; *Straub et al., 2009*), transcriptional remodeling through NFATc3 signaling (*Nieves-Cintrón et al., 2015*; *Nieves-Cintrón et al., 2018*; *Nilsson et al., 2006*; *Nystoriak et al., 2014*), and alterations in PKC/Rho kinase signaling (*Abd-Elrahman et al., 2015*; *Hien et al., 2016*; *Kizub et al., 2010*). Our group has additionally established, in human and mouse arterial tissue, PKA-mediated potentiation of LTCC activity via increased phosphorylation of the pore-forming $Ca_V1.2$ subunit at $Ser^{1928}$ as a key event that contributes to vascular complications during diabetic hyperglycemia (*Morotti et al., 2017*; *Navedo et al., 2010b*; *Nystoriak et al., 2017*). The classic molecular machinery stimulating PKA activity implicates activation of upstream effector enzymes (e.g. AC) and G protein-coupled receptors (GPCR). In the present study, using native human arteries and arterial myocytes, we identified a $G_s$-coupled P2Y receptor with the molecular, pharmacological, and signaling properties of $P2Y_{11}$ as the GPCR underlying the glucose triggering cascade that results in PKA-dependent potentiation of LTCCs (*Figure 8D*). Indeed, $P2Y_{11}$ is known to associate with ACs to stimulate cAMP/PKA signaling (*Communi et al., 1997*; *Pucéat et al., 1998*). Similar results were observed in mouse arteries and arterial myocytes, indicating that a $P2Y_{11}$-like receptor could be mediating the glucose response in these cells. The implications of these observations are profound, as they may shed light on unique mechanisms of diabetic vascular complications.

Elevations in extracellular glucose have been shown to promote autocrine release of nucleotides in a number of cells, including arterial myocytes (*Costa et al., 2009*; *Hazama et al., 1998*; *Nilsson et al., 2006*; *Parodi et al., 2002*). This glucose-induced release of nucleotides has been associated with increased global $[Ca^{2+}]_i$ in arterial myocytes, but the mechanisms and functional implications were not clearly established (*Nilsson et al., 2006*). This prompted us to examine whether extracellular nucleotides could be mediating glucose effects on LTCC activity and vascular

reactivity. Our first major finding was that hydrolysis of extracellular nucleotides by the ectonucleotidase apyrase prevented the increases in Ser$^{1928}$ phosphorylation, LTCC activity and vasoconstriction in response to elevations in extracellular glucose (*Figure 1*). Outcomes from LTCC recordings under continuous flow or static bath conditions (*Figure 1—figure supplement 2*) further support the involvement of extracellular nucleotide signaling in mediating the glucose effects. Given the well-established function of extracellular nucleotides in activating P2Y receptors (see (*von Kügelgen and Harden, 2011*)), our studies also pointed to the potential contribution of at least one of the eight known P2Y receptors in this process. In line with this concept, our multidisciplinary approach using human tissue suggested involvement of a P2Y receptor fitting the profile for the P2Y$_{11}$ subtype. Intriguingly, similar results were observed using mouse tissue/cells, suggesting a role for a P2Y$_{11}$-like receptor. To date, this is the only G$_s$-coupled P2Y, which could activate AC, leading to increased cAMP synthesis and thereby PKA activation. Considering the essential actions of PKA on LTCC activity and vasoconstriction in response to elevated glucose (*Morotti et al., 2017*; *Navedo et al., 2010b*; *Nystoriak et al., 2017*), we submit that such changes may proceed through activation of this purinergic signaling pathway.

Few studies have examined the role of P2Y$_{11}$ in the cardiovascular system. Current knowledge suggests that this receptor is associated with positive inotropic effects of ATP on cardiomyocytes and its activity seems to be impaired during cardiomyopathy (*Balogh et al., 2005*). Moreover, a polymorphism (Ala-87-Thr) in P2Y$_{11}$ has been linked to increased risk of acute myocardial infarction and C-reactive protein blood levels (*Amisten et al., 2007*). These results indicate a role for P2Y$_{11}$ in cardiac (dys)function. However, the functional significance of P2Y$_{11}$ in the vasculature, and particularly in arterial myocytes, is virtually unknown. In the present study, Western blot analysis detected an immunoreactive band of the expected molecular weight for P2Y$_{11}$ (e.g. ~40 ± 10 kDa) in human arterial lysates (*Figure 2C*). We confirmed robust distribution of the receptor at the expected plasma membrane location in isolated human arterial myocytes (*Figure 2D*). In line with the expected function for this GPCR, we corroborated localized subplasmalemmal cAMP synthesis upon receptor stimulation with a highly selective P2Y$_{11}$ agonist (NF546 (*Meis et al., 2010*); *Figure 5*). Notably, 20 mM D-glucose triggered cAMP synthesis of the same magnitude as the NF546 compound, and simultaneous application of NF546 and 20 mM D-glucose did not have any additional effect when compared with individual treatment exposure, suggesting that both stimuli proceed through the same signaling pathway. In support of this possibility, the specific P2Y$_{11}$ antagonist NF340 (*Meis et al., 2010*) blocked the NF546 and NF546 + 20 mM D-glucose effects on cAMP synthesis (*Figure 5*). The P2Y$_{11}$ antagonist also prevented increased Ser$^{1928}$ phosphorylation and LTCC activity in response to acute elevations in extracellular glucose (*Figure 6*), while the P2Y$_{11}$ agonist NF546 recapitulated all the glucose-induced, PKA-dependent effects (*Figure 7*). Moreover, the NF340 compound hampered glucose-induced changes in Ser$^{1928}$ phosphorylation levels and LTCC activity during chronic hyperglycemic conditions resembling those observed during diabetes (*Figure 8*). Altogether, these results revealed a role for human P2Y$_{11}$ or mouse P2Y$_{11}$-like receptor in arterial myocytes, particularly during glucose signaling, and suggest that this GPCR may be involved in the initiation and/or progression of arterial myocyte dysfunction leading to vascular complications during diabetic hyperglycemia. Our results may also have profound clinical implications as they add to a growing body of evidence implicating a role for altered P2Y$_{11}$ function in a number of pathological conditions (*Nishimura et al., 2017*), which could make them a potential therapeutic target.

Our findings indicating that elevated extracellular glucose triggers PKA-dependent potentiation of LTCC activity to induce vasoconstriction were unexpected, as PKA activity has been traditionally associated with vasodilation due, in part, to increased K$^+$ channel activity. A reasonable prediction was that glucose could influence LTCCs via a cAMP/PKA signaling pathway that is distinct from that governing K$^+$ channels as occurs upon β adrenergic stimulation (*Moore et al., 2014*). Such separation in cAMP would require precise subsarcolemmal compartmentalization of key proteins and signaling molecules (including GPCR, signaling enzyme, second messenger, effector protein and substrate). In the case of PKA-dependent effects on LTCC function in response to elevated glucose, this separation would include seclusion of a G$_s$-coupled receptor, AC, PKA and Ca$_V$1.2. Indeed, *Tajada et al* recently showed that optimal ion channel regulation by effector proteins is constrained by their nanometer distance between each other (*Tajada et al., 2017*). In support of this possibility, our recent work revealed nanometer proximity of a subpopulation of Ca$_V$1.2 (~10% of Ca$_V$1.2 clusters) to pools of PKA (*Nystoriak et al.,*

*2017*). Here, we found a close association ($\leq$40–90 nm) between pools of $P2Y_{11}$ and $Ca_V1.2$ as well as $P2Y_{11}$ and PKA (*Figure 3*). Moreover, given the cAMP biosensor data demonstrating that both glucose and the $P2Y_{11}$ agonist NF546 induce localized subsarcolemmal cAMP synthesis (*Figure 5*), it is also likely that $P2Y_{11}$ may be in close proximity to a specific AC isoform in arterial myocytes - the identity of which remains to be established. Nevertheless, this implicates the formation of a distinctive nanomolecular domain likely containing $P2Y_{11}$, AC, PKA and $Ca_V1.2$ in arterial myocytes (*Figure 8D*), although its confirmation is beyond the scope of this study. This nanodomain may ensure the necessary signaling compartmentalization required for PKA-mediated effects on LTCC activity and vascular reactivity in response to elevated glucose.

The spatial organization of the $P2Y_{11}$/AC/PKA/$Ca_V1.2$ nanodomain could be facilitated by scaffold proteins, such as A kinase anchoring proteins (AKAPs). These AKAPs fine tune signal transduction by localizing GPCRs, signaling enzymes and effector proteins in close proximity to their substrates (*Johnstone et al., 2018*; *Langeberg and Scott, 2015*). The A kinase anchoring protein 150 (AKAP150; murine ortholog of human AKAP79) is of particular interest as it has been shown to interact with AC, PKA and $Ca_V1.2$ (*Bauman et al., 2006*; *Efendiev et al., 2010*; *Gao et al., 1997*; *Murphy et al., 2014*; *Navedo et al., 2008*; *Nystoriak et al., 2017*; *Zhang et al., 2013*). Intriguingly, although AKAP79/150 can associate with multiple GPCRs (*Fraser et al., 2000*; *Zhang et al., 2011*), no report to date has linked it to any purinergic receptor. Localization of $P2Y_{11}$ within the same AKAP79/150-driven complex may be critical as this scaffold protein is necessary for increased L-type $Ca^{2+}$ channel activity in response to elevated glucose (*Nystoriak et al., 2017*). Two possibilities that do not exclude a fundamental role for AKAP79/150 are 1) direct interaction of $P2Y_{11}$ with the scaffold protein and/or 2) association of the receptor with a pool of $Ca_V1.2$ within the AKAP79/150 complex. Future studies should examine these exciting alternatives as they may provide further insight into mechanisms regulating LTCCs and vascular reactivity during diabetic hyperglycemia.

A lingering question is how nucleotides, such as ATP, are released in response to glucose to modulate arterial myocyte excitability. Although our data do not identify the specific nucleotide, ATP release is particularly interesting as this nucleotide is an effective vasoactive agent and the endogenous $P2Y_{11}$ agonist (*Communi et al., 1997*; *Kennedy, 2017*; *Schuchardt et al., 2012*; *von Kügelgen and Harden, 2011*). Indeed, ATP can activate $P2Y_{11}$ with an $EC_{50}$ that ranges from 1.8 to 17 µM, depending on the readout and the cell type used (*Communi et al., 1999*; *Haas et al., 2014*; *Meis et al., 2010*; *Qi et al., 2001*). Moreover, $P2Y_{11}$ can be activated by ATP/ATP-derived nucleotides with the following potency order: ATPγS > dATP > ATP>>>ADP, with other nucleotides such as UTP and UDP having minimal or no influence on receptor activation (*Kennedy, 2017*; *Schuchardt et al., 2012*; *von Kügelgen and Harden, 2011*). Our recent study suggested that glucose effects on LTCC activity and vascular reactivity require glucose transport into the cell and subsequent metabolism that can generate ATP (*Nystoriak et al., 2017*). ATP can then be transported out of the cell through ATP-binding cassettes, vesicular exocytosis, plasma membrane $F_1F_0$-ATPase, connexin hemichannels and pannexin channels to induce activation of $P2Y_{11}$ (*Lohman et al., 2012*). We thus speculate that one of these ATP transporters could form part of the AKAP/$P2Y_{11}$/AC/PKA/$Ca_V1.2$ nanocomplex to provide another layer of compartmentalization that facilitates PKA-dependent regulation of LTCC function and vascular reactivity in response to elevations in extracellular glucose.

In summary, our data in human and murine tissue provide evidence of a $G_s$-coupled purinergic receptor that fits the profile of $P2Y_{11}$ or a $P2Y_{11}$-like receptor, respectively, as a key upstream component in the signaling cascade regulating LTCC activity and vascular reactivity during diabetic hyperglycemia. Data start to define a unique nanomolecular complex likely formed by AKAP/$P2Y_{11}$/AC/PKA/$Ca_V1.2$. This nanocomplex can organize glucose signaling and help explain the intriguing consequences of glucose-induced PKA activation on LTCC function and vascular reactivity, which may have significant pathological implications, and therapeutic potential to treat vascular complications during diabetic hyperglycemia.

## Materials and methods

**Key resources table**

*Continued on next page*

*Continued*

| Reagent type (species) or resource | Designation | Source or reference | Identifiers | Additional information |
|---|---|---|---|---|
| Reagent type (species) or resource | Designation | Source or reference | Identifiers | Additional information |
| Strain (*Mus musculus*), C57BL/6J | wild-type | Jackson Laboratories | stock # 000664 | |
| Strain (*Mus musculus*) | S1928A | (*Lemke et al., 2008*) | | |
| Cell line (human embryonal kidney) | tsA-201 | Sigma-Aldrich | 96121229 | SV40 transformed |
| Oligode oxynucleotide, sense | P2Y$_{11}$ SNS ODN | Integrated DNA Technologies | human P2Y$_{11}$ sequence (NC_000019.10): 5'-CAACGTCTCGG GTAAGGAGAA-3' and 5'-ATGAGGAAGG AAACGTGGGT-3' | last three bases on the 3' end were phosphoro-thioated |
| Oligode oxynucleotide, antisense | P2Y$_{11}$ ANS ODN | Integrated DNA Technologies | human P2Y$_{11}$ sequence (NC_000019.10): 5'-CAAGGCCACCCTAACCACTG-3' and 5'-CTCTCCCTTCCCTGCGTTA-3' | last three bases on the 3' end were phosphoro-thioated |
| DNA construct | human P2Y$_{11}$ | UMR cDNA Resource Center | www.cDNA.org; clone ID P2Y1100000 | tagged with GFP at C-terminus |
| Antibody | anti-FP1 (Ca$_V$1.2; custom rabbit) | (*Davare et al., 2000*) | | dilutions: 1:100 for immunoblot and PLA; 10 μg/mL for GSD |
| Antibody | anti-CH3P (pSer$^{1928}$, custom rabbit) | (*Davare et al., 2000*) | | 1:50-1:100 dilutions |
| Antibody | anti-β-actin (mouse monoclonal) | Abcam | ab8226 RRID: AB_30637 | 1:1000 dilution |
| Antibody | anti-α-tubulin (mouse monoclonal) | Active Motif | 39527; clone 5-B-1–2 | 1:500 dilution |
| Antibody | anti-P2Y$_{11}$ (rabbit polyclonal) | Abcam | ab180739 | 1:100-1:200 dilutions |
| Antibody | anti-P2Y$_{11}$ (goat polyclonal) | Santa Cruz Biotechnology | sc-69588; clone C-18 RRID: AB_21559 | dilutions: 1:100 for stainings (PLA, classical) and 10 μg/mL for GSD |
| Antibody | anti-PKA$_{cat}$ α, β, γ (mouse monoclonal) | Santa Cruz Biotechnology | sc-28315; clone A-2 | 1:200 dilution |
| Antibody | PKA$_{cat}$ α, β, γ blocking peptide | Santa Cruz Biotechnology | sc-28315 P; clone A-2 | 1:20 dilution for 1 μg of primary antibody |
| Antibody | anti-PKA$_{cat}$ α, β, γ (rabbit polyclonal) | Santa Cruz Biotechnology | sc-28892; clone H-95 | dilutions: 1:200 for PLA and 10 μg/mL for GSD |
| Antibody | Alexa Fluor 488 conjugate of wheat germ agglutinin | Life Technologies | W7024 | |
| Antibody | Alexa Fluor 568-conjugated donkey anti-goat | Molecular Probes | A11057 RRID: AB_142581 | 5 mg/mL dilution |

*Continued on next page*

Continued

| Reagent type (species) or resource | Designation | Source or reference | Identifiers | Additional information |
|---|---|---|---|---|
| Antibody | Alexa Fluor 568-conjugated donkey anti-mouse | Molecular Probes | A10037 | 5 mg/mL dilution |
| Antibody | Alexa Fluor 568-conjugated donkey anti-rabbit | Molecular Probes | A11011 RRID: AB_143157 | 2 µg/mL dilution |
| Antibody | Alexa Fluor 647-conjugated donkey anti-goat | Molecular Probes | A21447 RRID: AB_141844 | 2 µg/mL dilution |
| Antibody | goat anti-rabbit IgG (H + L) -horseradish peroxidase conjugate | Bio-Rad | 170–6515 RRID: AB_11125142 | 1:10000 dilution |
| Antibody | goat anti-mouse IgG (H + L)- horseradish peroxidase conjugate | Bio-Rad | 170–6516 | 1:10000 dilution |
| Chemical compound, drug | sodium pentobarbital (Fatal-Plus) | Vortech Pharma-ceuticals | NDC 0298-9373-68 | |
| Chemical compound, drug | mannitol | Fisher Scientific | BP686 | |
| Chemical compound, drug | NF340 | Santa Cruz Biotechnology | sc-361274 | |
| Chemical compound, drug | NF546 | Tocris | 3892 | |
| Chemical compound, drug | apyrase | New England Biolabs | M0398L | |
| Chemical compound, drug | nifedipine | Sigma-Aldrich | N7634 | |
| Chemical compound, drug | Bay K-8644 | Sigma-Aldrich | 71145-03-4 | |
| Chemical compound, drug | forskolin | Sigma-Aldrich | F6886 | |
| Chemical compound, drug | amphotericin B | Sigma-Aldrich | A4888 | |
| Chemical compound, drug | MRS2578 | Santa Cruz Biotechnology | sc-204103A | |
| Chemical compound, drug | MRS2179 | Tocris | 1454889-37-2 | |
| Chemical compound, drug | protein kinase A inhbitor (PKI) | Sigma-Aldrich | P9115 | fragment 14–22, myristoylated trifluoroacetate salt |

*Continued*

| Reagent type (species) or resource | Designation | Source or reference | Identifiers | Additional information |
|---|---|---|---|---|
| Chemical compound, drug | Rp-Adenosine 3',5'-cyclic monophos-phorothioate triethylam-monium salt (rpcAMP) | Sigma-Aldrich | A165 | |
| Chemical compound, drug | Adenosine 5'-[γ-thio] triphosphate tetralithium salt (ATPγS) | Sigma-Aldrich | A1388 | |
| Software, algorithm | GraphPad Prism | | GraphPad Prism, RRID: SCR_002798 | |
| Software, algorithm | ImageJ | | Fiji, RRID: SCR_002285 | |
| Software, algorithm | pCLAMP10 | Molecular Devices | | electrophysiology |
| Software, algorithm | LASAF | Leica | | GSD |
| Software, algorithm | IonOptix | IonOptix | | arterial diameter recordings |
| Software, algorithm | Metaflor | Molecular Devices | | FRET |

## Animals

Male wild type C57BL/6J (wt) or knockin mice in which Ser$^{1928}$ of Ca$_V$1.2 was mutated to Ala (S1928A) (*Lemke et al., 2008*) of 5 to 8 weeks of age were euthanized with a lethal dose of sodium pentobarbital (250 mg/kg; intraperitoneally), as approved by the University of California, Davis Animal Care and Use Committee (protocol #: 20321).

## Human tissue

Excised adipose arteries from human patients undergoing surgical sleeve gastrectomy were used for this study (*Supplementary file 3*). Samples were obtained after Institutional Review Board (IRB) approval from the University of Nevada Reno School of Medicine (IRB ID: 2013–019) and in accordance with the guidelines of the *Declaration of Helsinki*. The need for informed consent was waived by IRBs at the University of Nevada Reno School of Medicine (IRB ID: 2013–019) and the University of California Davis (IRB ID: 597267–1) because the tissue is considered 'waste', has no codification that could be used to identify patients and was determined not to be human subject research in accordance with United States of America federal regulations, as defined by 45 CFR 46.102(f). This precludes the acquisition of detailed clinical profiles other than sex, age and whether the patient is nondiabetic or diabetic. Only samples from nondiabetic patients were used. No exclusions were made due to medication history, sex or presence of comorbidities. Collected tissue was placed in cold phosphate-buffered saline (PBS) solution containing (in mM): 138 NaCl, 3 KCl, 10 Na$_2$HPO$_4$, 2 NaH$_2$PO$_4$, 5 D-glucose, 0.1 CaCl$_2$, and 0.1 MgSO$_4$, pH 7.4 with NaOH until used.

## Arterial myocyte isolation

Mouse cerebral arteries were dissected in ice-cold dissection buffer containing (in mM): 140 NaCl, 5 KCl, 2 MgCl$_2$, 10 D-glucose, 10 HEPES, pH 7.4, with NaOH. Arteries were digested in dissection buffer containing papain (1 mg/mL) and dithiothreitol (1 mg/mL) at 37° C for 7 min, followed by incubation in dissection buffer containing collagenase type F (0.7 mg/mL) and collagenase type H (0.3 mg/mL) at 37° C for 7 min. Arteries were washed in ice-cold dissection buffer and gently triturated with glass pipettes to disperse the cells, which were kept in ice-cold dissection buffer until use.

Single human arterial myocytes (*Supplementary file 3*) were isolated as previously described (*Nieves-Cintrón et al., 2017*; *Nystoriak et al., 2017*). Briefly, small diameter adipose arteries from human samples were enzymatically digested in dissection solution supplemented with papain (26 U/mL) and dithiothreitol (1 mg/mL) at 37° C for 15 min, followed by incubation in dissection solution containing collagenase type H (1.95 U/mL), elastase (0.5 mg/mL), and trypsin inhibitor (1 mg/mL) at 37° C for 10 min. Cells were then washed three times in ice-cold dissection buffer, triturated with glass pipettes to disperse the individual cells, and maintained in ice-cold dissection buffer until use.

For unpassaged, cultured human and mouse arterial myocytes, human adipose arteries and mouse aortas were dissected out and placed in ice-cold Dulbecco's Modified Eagle Medium (DMEM; Gibco – Life Technologies, Grand Island, NY) containing 1X glutamate, 1X pyruvate, 1X penicillin/streptavidin and fungizone (0.25 g/mL). Artery segments were subsequently transferred and incubated in a DMEM solution containing 2.2 mg/mL of collagenase Type 2 (Worthington) at 37° C for 15 min to remove the adventitia. To disperse and culture unpassaged arterial myocytes, the tissue was cut into 2–5 mm segments and incubated at 37° C with constant shaking in a buffer containing (in mM): 134 NaCl, 6 KCl, $CaCl_2$, 10 HEPES, and 7 D-glucose supplemented with 2.2 mg/mL collagenase Type 2 (Worthington). The digestion was stopped by adding an equal volume of DMEM containing 5% fetal bovine serum. Digested tissue was then centrifuged for 5 min at 14,000 rpm. The supernatant was removed and the pellet containing the digested tissue was resuspended in DMEM containing 1X glutamate, 1X pyruvate, 5% serum and 5 mM D-glucose, resulting in dispersion of individual arterial myocytes. Cells were then seeded on glass coverslips coated with laminin and kept in an incubator at 37° C with 5% $CO_2$ for 2–3 days before adenoviral transduction or lysis for immunoblotting (see section below).

## Arterial diameter measurements

Freshly isolated posterior mouse cerebral arteries were cannulated on glass micropipettes mounted in a 5 mL myograph chamber (University of Vermont Instrumentation and Model Facility), as described previously (*Nieves-Cintrón et al., 2017*; *Nystoriak et al., 2014*; *Nystoriak et al., 2017*). Arteries were allowed to equilibrate at an intravascular pressure of 20 mmHg, while being continuously superfused (37° C, 30 min, 3–5 mL/min) with physiological saline solution composed of (in mM): 119 NaCl, 4.7 KCl, 2 $CaCl_2$, 24 $NaHCO_3$, 1.2 $KH_2PO_4$, 1.2 $MgSO_4$, 0.023 EDTA, and 10 D-glucose aerated with 5% $CO_2$/95% $O_2$. Bath pH was closely monitored and maintained at 7.35 to 7.4. After equilibration, intravascular pressure was increased to 60 mmHg and arteries allowed to develop stable myogenic tone. Arteries not exhibiting stable tone after ~1 hr were discarded. To assess the response of arterial diameter to hyperglycemia, D-glucose was increased from 10 mM to 20 mM in the perfusion solution. Arterial tone data are presented as a percent decrease in diameter relative to the maximum passive diameter at 60 mmHg obtained using $Ca^{2+}$-free saline solution containing nifedipine (1 µM) at the end of the experiment.

## Electrophysiology

All experiments were performed at room temperature (22–25° C). Both whole-cell and single-channel data were acquired using an Axopatch 200B amplifier and Digidata 1440 digitizer (Molecular Devices). Recording electrodes were pulled from borosilicate capillary glass using a micropipette puller (model P-97, Sutter Instruments) and subsequently polished to achieve resistances that ranged from 3.5 to 6.5 MΩ. L-type $Ca^{2+}$ channel currents were assessed in freshly dissociated arterial myocytes using the perforated whole-cell mode of the patch-clamp technique with $Ba^{2+}$ as a charge carrier ($I_{Ba}$) after myocytes were allowed to attach (10 min) to a glass coverslip in a recording chamber. Borosilicate glass pipettes were filled with a solution containing (in mM): 120 CsCl, 20 tetraethylammonium chloride (TEA-Cl), 1 EGTA, and 20 HEPES with amphotericin B (250 µg/mL; pH adjusted to 7.2 with CsOH). The bath solution consisted of (in mM): 115 NaCl, 10 TEA-Cl, 0.5 $MgCl_2$, 10 D-glucose, 5 CsCl, 20 $BaCl_2$, and 20 HEPES, pH adjusted to 7.4 with CsOH. $I_{Ba}$ were elicited by 200 ms depolarizing pulses from a holding potential of −70 mV to +10 mV. $I_{Ba}$ were first recorded in the presence of 10 mM D-glucose under static flow conditions. This solution was then exchanged for a solution containing either 20 mM D-glucose, 20 mM mannitol or NF546 at a rate of 2.1 mL/min. Flow was stopped after 3 min, and $I_{Ba}$ were recorded again under static flow conditions at least 5 min after the indicated treatment was initiated. In some experiments, arterial myocytes were

pretreated for 10–15 min with indicated inhibitors prior to $I_{Ba}$ recordings. For experiments comparing $I_{Ba}$ before and after elevated glucose under continuous flow and static bath conditions, cells were patched in a bath solution containing 10 mM D-glucose at a perfusion rate of 2.1 mL/min. After establishment of a stable gigaseal for at least 5 min, $I_{Ba}$ were recorded in the presence of 10 mM D-glucose under continuous flow. Those cells that showed robust $I_{Ba}$ in 10 mM D-glucose were then perfused with a bath solution containing 20 mM D-glucose under continuous flow for at least 5 min before recording of $I_{Ba}$ again. Subsequently, $I_{Ba}$ were recorded one more time on the same cell, 5 min after stopping the bath perfusion (e.g. static bath conditions). Currents were sampled at 10 kHz and low pass-filtered at 5 kHz. The $Ca_V1.2$ blocker nifedipine (1 µM) was applied at the end of each experiment to determine the nifedipine-sensitive component. The I-V relationship of nifedipine-sensitive $I_{Ba}$ was determined by 200 ms depolarizing steps from −70 mV to voltages ranging from −60 to +60 mV in increments of +10 mV. The I-V relationship for averaged data sets was fit with a peak Gaussian function: $I(V)=I_{max} \times \exp(-0.5((V - V_{max})/b)^2)$, where $I_{max}$ is peak $I$, $V_{max}$ is $V$ at $I_{max}$, and $b$ is the slope of the distribution, as described previously (*Nystoriak et al., 2017*). A voltage error of 10 mV attributable to the liquid junction potential of the recording solutions was corrected offline.

The cell-attached configuration of the patch-clamp technique was used to examine single-channel $Ca^{2+}$ currents in arterial myocytes using $Ca^{2+}$ as the charge carrier, as previously described (*Cheng et al., 2011*; *Dixon et al., 2015*; *Dixon et al., 2012*; *Navedo et al., 2010a*; *Nystoriak et al., 2017*). Resting membrane potential was fixed to ~0 mV using a high $K^+$ bath solution (*Hess et al., 1986*; *Zampini et al., 2006*) composed of (in mM): 145 KCl, 10 NaCl, and 10 HEPES, pH adjusted to 7.4 with NaOH. Pipette solution was composed of (in mM): 120 TEA-Cl, 110 $CaCl_2$, and 10 HEPES with BayK-8644 (500 nM), which promotes longer open times of the channel thereby increasing the probability of sweeps with channel activity. Single L-type $Ca^{2+}$ currents were elicited by a 2 s depolarizing step from a holding of −70 mV to −30 mV. Single-channel currents were sampled at 10 kHz and low pass filtered at 2 kHz, followed by a Gaussian filter (500 Hz) during analysis. Capacitive currents were compensated by subtraction of blank sweeps. When nifedipine was included in the patch pipette, $Ca_V1.2$ activity was negligible. Single-channel openings and $nP_o$ were analyzed using the single-channel event half-amplitude detection algorithm from pCLAMP 10.

## Immunoprecipitation and immunoblotting

Human adipose arteries, mouse cerebral and mesenteric arteries, and mouse arterial myocytes were homogenized in a RIPA lysis buffer solution composed of 50 mM Tris-HCl (pH 7.4), 150 mM NaCl, 5 mM EGTA, 10 mM EDTA, 1% nonyl phenoxypolyethoxylethanol-40 (NP-40), 10% glycerol, 0.05% sodium dodecyl sulfate (SDS), 0.4% deoxycholic acid (DOC) with protease inhibitors (1 µg/ml pepstatin A, 10 µg/ml leupeptin, 20 µg/mL aprotinin, and 200 nM phenylmethylsulfonyl fluoride) and phosphatase inhibitors (2 µM microcystin LR, 1 mM $p$-nitrophenyl phosphate, 50 mM Na-pyrophosphate, and 50 mM NaF), then sonicated for 5 min at 4° C and cleared by centrifugation (15,000 X $g$) for 20 min at 4° C. For $Ser^{1928}$ phosphorylation experiments, the soluble fraction was incubated on a head-over-head tilter with protein A-Sepharose beads and a purified custom antibody against $Ca_V1.2$ (5 µg of FP1; see *Davare et al., 2000*) or nonspecific rabbit immunoglobulin G (IgG) control for 4 hr at 4° C. Beads were washed three times with washing buffer containing (in mM): 150 NaCl, 10 EDTA, 10 EGTA, 10 Tris-HCl (pH 7.4), 0.1% Triton X-100, pH 7.4. Samples were extracted in Laemmli Sample Buffer (Bio-Rad) for 5 min at 95° C for immunoprecipitation ($Ser^{1928}$ phosphorylation) experiments and 15 min at 80° C for other immunoblot experiments. Proteins were separated by SDS-polyacrylamide gel electrophoresis at 75–100 V for 1.5 hr in a stacking gel polymerized from 3% acrylamide and a resolving phase polymerized from 7.5% ($Ser^{1928}$ phosphorylation) or 10% acrylamide. Proteins were then transferred to a polyvinylidene difluoride membrane at 50 V for 600 min at 4° C. All membranes, except for $P2Y_{11}$ blots (10% Odyssey blocking buffer; LI-COR Biosciences), were blocked in 5% nonfat dried milk in tris-buffered saline with 0.05% Tween 20 (TBS-T) for 1 hr at room temperature before primary antibody incubation for 2–3 hr at room temperature. Antibodies and dilutions were as follows: rabbit anti-CH3P ($pSer^{1928}$; 1:50-1:100),(*Davare et al., 2000*) rabbit anti-FP1 ($Ca_V1.2$; 1:100) (*Davare et al., 2000*), rabbit anti-$P2Y_{11}$ (Abcam, 1:100-1:200); mouse anti-β-actin (Abcam, 1:1000), and anti α-tubulin (Active Motif, 1:500). Except for the $P2Y_{11}$ antibody, which was diluted in 1% bovine serum albumin, all antibodies were diluted in 5% nonfat dried milk in TBS-T. After washing, membranes were then incubated 1 hr at room temperature with horseradish peroxidase-labeled goat anti-rabbit (Bio-Rad; 1:10,000) or goat-anti-mouse antibodies (Bio-Rad; 1:10,000)

diluted in 5% nonfat dried milk in TBS-T or 5% Odyssey in TBS-T (P2Y$_{11}$ only) and developed on autoradiography film using chemiluminescence (Classico [Millipore] and Femto [Thermo Fisher Scientific]). For Ser$^{1928}$ phosphorylation and P2Y$_{11}$ knock down experiments, total Ca$_V$1.2 and β-actin, respectively, were used for normalization (density expressed as percentage of total Ca$_V$1.2/β-actin and these ratios were normalized to the ratio of the control band to get the relative density of pSer$^{1928}$/Ca$_V$1.2 or P2Y$_{11}$/β-actin for all treatments). For Ser$^{1928}$ phosphorylation quantification in *Figure 1—figure supplement 1E*, triplicates were run in parallel and all values were normalized to the ratio of the loading control band (e.g. α-tubulin obtained from the same samples) to get the pSer$^{1928}$/α-tubulin ratio for all treatments. The pSer$^{1928}$/α-tubulin ratios were then normalized to the first control band ratio to obtain the relative density of pSer$^{1928}$/α-tubulin. Densitometry analysis for bands was performed with ImageJ software (National Institutes of Health) as follow. Developed films were scanned to tiff files, uploaded to ImageJ and color inverted. Immunoreactive bands were outlined and light intensity per area was measured. Doublet bands were outlined and measured as one signal. Intensity of equal area above and below each immunoreactive band was measured to average and then subtract the background signal. Importantly, films were exposed for increasing time periods to ensure signals were in the linear range as described in *Davare and Hell (2003)* and *Hall et al. (2006)*. β-actin and α-tubulin were used as a loading control in some experiments.

## Organ culture

Freshly isolated arteries for acute Ser$^{1928}$ phosphorylation experiments were incubated for 5 min at room temperature in a physiological saline solution composed of (in mM): 119 NaCl, 4.7 KCl, 2 CaCl$_2$, 24 NaHCO$_3$, 1.2 KH$_2$PO$_4$, 1.2 MgSO$_4$, 0.023 EDTA, and 10 (5 for humans) D-glucose aerated with 5% CO$_2$/95% O$_2$ to reach a pH of 7.35–7.4. Arteries were then incubated for 10 min at room temperature in a similar physiological solution containing the specified treatment. Following incubation, arteries were immediately placed in RIPA lysis buffer with protease and phosphatase inhibitors (as described in immunoblotting section). For chronic exposure experiments, arteries were placed in DMEM/F-12 (Gibco-Life Technologies, Grand Island, NY) supplemented with L-glutamine (2 mM) containing 10 mM, 20 mM D-glucose or 20 mM mannitol in the absence or presence of 10 μM NF340 and incubated for 48 hr at 37° C. After incubation, arteries were quickly washed in PBS and then placed in RIPA lysis buffer with protease and phosphatase inhibitors for immunoblotting.

## Immunofluorescence

Immunofluorescent labeling of freshly dissociated arterial myocytes was performed as described previously (*Navedo et al., 2008*), using a goat-anti P2Y$_{11}$ (Santa Cruz Biotechnology, clone C-18; 1:100) or a mouse-anti PKA$_{cat}$ α, β, γ (Santa Cruz Biotechnology, clone A-2, 1:200) antibodies in PBS supplemented with 0.1% BSA. Alexa Fluor 488 conjugated wheat germ agglutinin (WGA; Life Technologies) was used to stain for the plasma membrane. The secondary antibodies were Alexa Fluor 568-conjugated donkey anti-goat and donkey anti-mouse (5 mg/mL; Molecular Probes). In control experiments, the PKA$_{cat}$ antibody was preabsorbed with a PKA$_{cat}$ blocking peptide (Santa Cruz Biotechnology, A-2; 1:20 for 1 μg of primary antibody). Cells were imaged (512 × 512 pixel images) using an Olympus FV1000 confocal microscope paired with an Olympus 60x oil immersion lens (NA = 1.4) and a zoom of 3.0 (pixel size = 0.1 μm). P2Y$_{11}$-associated fluorescence was not detected in negative control experiments in which the primary antibody was substituted for PBS or boiled to test for antibody specificity. Cells for each group were imaged with the same laser power, gain settings, and pinhole.

## Proximity ligation assay

A Duolink In Situ PLA kit (Sigma) (*Fredriksson et al., 2002*) was used to detect complexes consisting of P2Y$_{11}$ and Ca$_V$1.2 and P2Y$_{11}$ and PKA$_{cat}$ in freshly isolated arterial myocytes as previously described (*Nieves-Cintrón et al., 2017*; *Nystoriak et al., 2017*). Briefly, cells were plated on glass coverslips and allowed to adhere (1 hr, room temperature) prior to fixing with 4% paraformaldehyde (20 min), quenching in 100 mM glycine (15 min), and washing in PBS (2 × 3 min). Cells were permeabilized with 0.1% Triton X-100 (20 min) and then blocked (1 hr, 37° C) in 50% Odyssey blocking solution (LI-COR Bioscience). Cells were incubated overnight at 4° C with a specific combination of two primary antibodies in 0.1% Odyssey +0.05% Triton X-100 PBS solution: goat anti-P2Y$_{11}$ (Santa Cruz

Biotechnology, clone C-18; 1:100), custom rabbit anti-FP1 (Ca$_V$1.2, 1:100) (*Davare et al., 2000*) and rabbit anti-PKA$_{cat}$ α, β, γ (Santa Cruz Biotechnology, clone H-95, 1:200). Cells were incubated with only one primary antibody as negative controls. After primary antibody incubation, cells were washed with Duolink buffer A (2 × 5 min). Oligonucleotide-conjugated secondary antibodies (PLA probes: anti-goat MINUS and anti-rabbit PLUS) were used to detect P2Y$_{11}$ and Ca$_V$1.2 and P2Y$_{11}$ and PKA$_{cat}$ (1 hr, 37° C). Following incubation with probes, cells were washed with Duolink buffer A (2 × 5 min) and a ligation solution composed of two distinct oligonucleotides, complementary to the probes, and ligase was added (30 min, 37° C) to allow hybridization with the probes and formation of a closed DNA circle at sites of dual labeling, which serves as a template for a rolling circle amplification reaction (100 min, 37° C). After the amplification step, cells were washed with Duolink buffer B (2 × 10 min) and 1% buffer B (1 × 1 min). Coverslips were allowed to dry and subsequently mounted on a microscope slide with Duolink mounting medium. The fluorescence signal was detected using an Olympus FV1000 confocal microscope coupled with a 60x oil immersion lens (NA, 1.4). Images were collected at different optical planes (z-axis step = 0.5 µm). The stack of images for each sample was combined into a single-intensity projection image that was subsequently used for analysis of number of puncta/µm$^2$ per cell.

## Immunolabeling and Ground State Depletion (GSD) microscopy

Isolated arterial myocytes were allowed to adhere to a coverslip (1 hr) before fixing with 3% paraformaldehyde +0.1% glutaraldehyde solution in PBS (10 min) followed by washes with PBS (3 × 15 min). Cells were then incubated for 5 min with 0.1% sodium borohydride in H$_2$O, followed by 3 × 5 min washes with PBS and incubated in permeabilization/blocking solution consistent of 0.05% Triton X-100% and 20% SEA BLOCK (Thermo Scientific) for 1 hr at room temperature. Cells were exposed overnight to primary antibodies [goat anti-P2Y$_{11}$ (Santa Cruz Biotechnology, clone C-18), custom rabbit anti-Ca$_V$1.2 (FP1 (*Davare et al., 2000*)) or rabbit anti-PKA$_{cat}$ α, β, γ (Santa Cruz Biotechnology, clone H-95)] diluted in blocking buffer to a concentration of 10 µg/mL. Cells were briefly washed 3x with PBS, followed by additional longer washes (3x for 15 min). For secondary antibodies, donkey Alexa Fluor 647-conjugated antibody recognizing goat IgG (2 µg/mL; Molecular Probes) and donkey Alexa Fluor 568-conjugated antibody recognizing rabbit IgG (2 µg/mL; Molecular Probes) diluted in blocking buffer were added for 1 hr at room temperature. Secondary Ab was washed 3x with PBS, followed by longer washes (3x for 15 min). Specificity of secondary antibodies was tested in control experiments in which primary antibodies were omitted from the preparation (no 1° antibody controls). For imaging, coverslips were mounted on microscope slides with a round cavity containing MEA-GLOX imaging buffer (NeoLab Migge Laborbedarf-Vertriebs GmbH, Germany) and sealed with Twinsil (Picodent, Germany). The imaging buffer was composed of 10 mM MEA (cysteamine), 0.56 mg/mL glucose oxidase, 34 µg/mL catalase, and buffer containing 10% w/v glucose, 10 mM NaCl, and 50 mM Tris-HCl, pH 8.

Super-resolution images of arterial myocytes were obtained using a super resolution ground state depletion system (SR-GSD, Leica) dependent on stochastic single-molecule localization, equipped with high-power lasers (532 nm, 2.1 kW/ cm$^2$; 642 nm, 2.1 kW/ cm$^2$) and an additional 30 mW, 405 nm laser. A 160x HCX Plan-Apochromat (NA 1.47) oil immersion lens and an electron-multiplying charge-coupled device (EMCCD) camera (iXon3 897; Andor Technology) were used to acquire images (*Dixon et al., 2015*). The camera was running in frame-transfer mode at a frame rate of 100 Hz (10 ms exposure time). Fluorescence was detected through Leica high-power TIRF filter cubes (532 HP-T, 642 HP-T) with emission band-pass filters of 550–650 nm and 660–760 nm. Reconstruction of P2Y$_{11}$, Ca$_V$1.2, and PKA distribution from 30,000 images used the coordinates of centroids obtained by fitting single-molecule fluorescence signals with a 2D Gaussian function using a LASAF software (Leica). The localization accuracy of the system is limited by the statistical noise of photon counting; the precision of localization is proportional to DLR/$\sqrt{N}$, where DLR is the diffraction-limited resolution of a fluorophore and N is the average number of detected photons per switching event, assuming the point-spread functions are Gaussian (*Dempsey et al., 2011*; *Fölling et al., 2008*). The full width at half maximum for single-molecule signals was ~20 nm as recently calculated by our group (*Tajada et al., 2017*). Localizations produced from less than 800 photons were filtered out of the reconstruction. All pixels with intensity above a user-defined threshold were binarized, evaluated and segmented into individual objects and included as clusters in our analysis. Cluster size

and density were determined using the Analyze Particle option in the ImageJ software (National Institute of Health). The JACoP plug-in in the ImageJ software was used to unbiasedly and automatically determine the shortest intermolecular distances for $P2Y_{11}$ and $Ca_V1.2$ as well as $P2Y_{11}$ and PKA-cat following the protocol described by *Bolte and Cordelières (2006)*. Intermolecular distance histograms were generated from the JACoP plug-in output data and fitted with a sum of two Gaussian functions of the following equation: $Y = Y_0 + (A_1 / (w_1 \times \sqrt{(\pi/2)})) \times \exp(-2 \times ((X - X_{C1})/w_1)^2) + (A_2 / (w_2 \times \sqrt{(\pi/2)})) \times \exp(-2 \times ((X - X_{C2})/w_2)^2)$, where $Y_0$ is the Y offset, $A_1$ and $A_2$ are the areas of the distribution of distances, $X_{C1}$ and $X_{C2}$ are the x values of distance at the center of the distribution, and $w_1$ and $w_2$ are the widths of each distribution in nanometers. To calculate the percentage of complete $P2Y_{11}$ overlap, we first quantified the number of clusters/objects in the thresholded, binarized localization maps for $P2Y_{11}$ and $Ca_V1.2$ as well as $P2Y_{11}$ and $PKA_{cat}$ using the Analyze Particle tool in ImageJ. The binarized localization maps of $P2Y_{11}$ were then multiplied by those corresponding to the paired $Ca_V1.2$ or $PKA_{cat}$ localization maps. The number of objects obtained from this operation were then divided by the number of clusters/objects present in the original binarized localization maps for $P2Y_{11}$. This method only detects objects that are 100% overlapping with each other in two independent images (e.g localization maps for $P2Y_{11}$ and $Ca_V1.2$). Random simulation of image pairs for $P2Y_{11}$ and $Ca_V1.2$ or $P2Y_{11}$ and $PKA_{cat}$ were generated based on the original super-resolution localization maps for each pair of proteins using the Coste's randomization algorithm included in the JACoP plug-in in ImageJ (*Bolte and Cordelières, 2006*) from six different cells per condition. Parameters for the Coste's randomization algorithm were selected to generate simulated images with relative similar cluster size and density to those observed in the original super-resolution localization maps for each protein. Each randomized image was generated after undergoing 1000 randomization rounds. Randomized images were binarized prior to analysis and the percentage of complete $P2Y_{11}$ overlap from the randomized image pairs was calculated as described above.

## Cell culture, transfection of tsA-201 cells, and reverse permeabilization

tsA-201 were obtained from Sigma-Aldrich (cat#: ECACC 96121229). These cells are included in the European Collection of Authenticated Cell Cultures. tsA-201 cells were cultured in DMEM supplemented with 1X pyruvate, 1X glutamax, 8% fetal bovine serum (FBS) and 5 mM glucose (without phenol red) at 37° C in a 5% $CO_2$ incubator. Cells were transfected at 60–70% confluence with human $P2Y_{11}$ tagged with GFP at C-terminus (UMR cDNA Resource Center - www.cDNA.org; clone ID P2Y1100000) with JetPRIME transfection reagent (Polyplus transfection SA, NY) for approximately 36 hr. $P2Y_{11}$ sense and antisense oligodeoxynucleotides (ODNs; 2 µM; Integrated DNA Technologies) were transfected in tsA-201 cells 24 hr after initial transfection with $P2Y_{11}$-GFP. The cells were gently washed with 1X PBS and then removed with a cell scrapper using a RIPA lysis buffer solution composed of 50 mM Tris base, 150 mM NaCl, 5 mM EGTA, 10 mM EDTA, 1% nonyl phenoxypolyethoxylethanol-40 (NP-40), 10% glycerol, 0.05% sodium dodecyl sulfate (SDS), 0.4% deoxycholic acid (DOC) with protease inhibitors (1 µg/mL pepstatin A, 10 µg/mL leupeptin, and 20 µg/mL aprotinin). Antisense and sense ODNs for $P2Y_{11}$ were designed and checked for specificity using Prime BLAST against the human $P2Y_{11}$ sequence (NC_000019.10). The sequences used for the antisense ODNs were as follows: 5'-CAAGGCCACCCTAACCACTG-3' and 5'-CTCTCCCTTCCCTGCGTTA-3'. The sequences for sense ODNs were 5'-CAACGTCTCGGGTAAGGAGAA-3' and 5'-ATGAGGAAG-GAAACGTGGGT-3'. For all ODNs, the last three bases on the 3' end were phosphorothioated to reduce degradation by cellular nucleases. ODNs were dissolved in nuclease-free water to a 1 mM concentration.

Human adipose arteries were permeabilized in a solution containing (in mM): 120 KCl, 2 $MgCl_2$, 10 EGTA, 5 $Na_2ATP$, 20 TES (2-[(2-Hydroxy-1, 1-bis(hydroxymethyl)ethyl)amino] ethanesulfonic acid, N-[Tris(hydroxymethyl)methyl]−2-aminoethanesufonic acid), pH adjusted to 6.8 with NaOH. Arteries were first incubated in permeabilization solution for 20 min at 4°C, followed by a 4 hr incubation at 4°C in a solution supplemented with ODNs (2 µM). Arteries were placed in an ODN-containing solution with elevated $MgCl_2$ (10 mM) for 1 hr at room temperature. Permeabilization was reversed by incubating arteries for 30 min at room temperature in MOPS physiological solution composed of (in mM): 140 NaCl, 5 KCl, 10 $MgCl_2$, 5 D-glucose, and 2 MOPS, pH adjusted to 7.1 with NaOH. Arteries were incubated in a MOPS solution containing 0.01, 0.1, and 1.8 mM $CaCl_2$ in 15 min intervals, increasing $Ca^{2+}$ gradually. After completing reverse permeabilization, arteries were cultured in

D-MEM/F-12 culture media supplemented with L-glutamine (2 mM) for 2.5 days at 37°C. Arteries were then lysed and used for Western blot (see Immunoblotting section).

### Adenovirus infection of unpassaged arterial myocytes and Fluorescence Resonance Energy Transfer (FRET)

Laminin (Life Technologies, Grand Island, NY) diluted 100x in sterile-filtered PBS (137 mM NaCl, 2.7 mM KCl, 10 mM $Na_2HPO_4$, 1.8 mM $KH_2PO_4$, pH = 7.4) was used to coat #0 glass coverslips (Karl Hecht, Sondheim, Germany). After adding diluted laminin (100 µL per coverslip), coverslips were placed in a 37° C incubator with 5% $CO_2$ for a minimum of 2 hr, then moved to 24-well plate wells (Falcon, Tewksbury, MA), and washed 3x with sterile-filtered PBS. Freshly dissociated human adipose arterial myocytes and mouse aortic cells were plated on the laminin-coated coverslips with 500 µL of serum-containing media for 48 hr in a 37° C incubator with 5% $CO_2$. Media was then replaced with 500 µL of serum-free media-containing virus coding for the membrane-targeted Epac1-camps-based FRET sensor (ICUE3-PM) (*Allen et al., 2006*; *Liu et al., 2011*) and placed at 37° C with 5% $CO_2$ for another 36 hr. Viruses were produced using the AdEasy system (Qbiogene, Carlsbad, CA) (*Luo et al., 2007*). After infection, media was changed to serum-free media without virus. Glass coverslips were transferred to glass bottom culture dishes (MatTek, Ashland, MA) containing 3 mL PBS at room temperature.

A Zeiss AXIO Observer A1 inverted fluorescence microscope (San Diego, CA) equipped with a Hamamatsu Orca-Flash 4.0 digital camera (Bridgewater, NJ) and controlled by Metaflor software (Molecular Devices, Sunnyvale, CA) acquired phase contrast, CFP480, and FRET images. Phase contrast and CFP480 images were collected with 20x and 40x oil immersion objective lenses, while FRET images were collected using only the 40x oil immersion objective lens. Images for FRET analysis were recorded by exciting the donor fluorophore at 430–455 nm and measuring emission fluorescence with two filters (475DF40 for cyan and 535DF25 for yellow). Images were subjected to background subtraction and acquired every 30 s with exposure time of 200 ms for each channel. The donor/acceptor FRET ratio was calculated and normalized to the ratio value of baseline. CFP480 images were acquired by exciting the donor fluorophore at 430–455 nm and measuring emission fluorescence with the 475DF40 filter for 25 ms. Averages of normalized curves and maximal response to stimulation were graphed based on FRET ratio changes. Binding of cAMP to ICUE3-PM led to decreases in the YFP/CFP ratio, indicating increases in cAMP levels.

### Chemicals and statistics

All chemical reagents were from Sigma-Aldrich (St. Louis, MO) unless otherwise stated. Data were analyzed using GraphPad Prism software and expressed as mean ±SEM. Data were assessed for potential outliers using the GraphPad Prism Outlier Test and for normality of distribution using appropriate tests. Statistical significance was then determined using suitable paired or unpaired Student's *t*-test, nonparametric tests or One-way analysis of variance (ANOVA) for multiple comparisons with proper post hoc test. $p < 0.05$ was considered statistically significant (denoted by * in figures).

## Acknowledgements

We thank members of the Navedo Lab for technical support. We also thank Dr. Charles Kennedy from the University of Strathclyde for helpful discussions and advice on purinergic signaling in the vasculature.

## Additional information

### Funding

| Funder | Grant reference number | Author |
| --- | --- | --- |
| National Heart, Lung, and Blood Institute | R01HL098200 | Manuel F Navedo |
| National Heart, Lung, and Blood Institute | R01HL121059 | Manuel F Navedo |

| | | |
|---|---|---|
| National Institute of Neurological Disorders and Stroke | R01NS078792 | Johannes W Hell |
| National Institute on Aging | R01AG055357 | Johannes W Hell |
| National Institute of General Medical Sciences | T32GM099608 | Maria Paz Prada |
| National Heart, Lung, and Blood Institute | T32HL086350 | Arsalan U Syed Matthew A Nystoriak |
| National Institute of Diabetes and Digestive and Kidney Diseases | R01DK57236 | Sean M Ward |
| National Institute of Neurological Disorders and Stroke | F31NS086226 | Olivia R Buonarati |
| American Heart Association | 16SDG27260070 | Matthew A Nystoriak |
| American Heart Association | 18POST34060234 | Debapriya Ghosh |
| National Heart, Lung, and Blood Institute | R01HL127764 | Yang K Xiang |
| National Heart, Lung, and Blood Institute | R01HL112413 | Yang K Xiang |
| University of California, Davis | Academic Federation Innovative Development Award | Madeline Nieves-Cintrón |

The funders had no role in study design, data collection and interpretation, or the decision to submit the work for publication.

## Author contributions

Maria Paz Prada, Conceptualization, Data curation, Formal analysis, Validation, Investigation, Visualization, Methodology, Writing—orignal draft, Writing—review and editing; Arsalan U Syed, Olivia R Buonarati, Debapriya Ghosh, Formal analysis, Validation, Investigation, Methodology, Writing—review and editing; Gopireddy R Reddy, Formal analysis, Validation, Investigation, Writing—review and editing; Matthew A Nystoriak, Formal analysis, Investigation, Writing—review and editing; Sergi Simó, Daisuke Sato, Formal analysis, Investigation, Methodology, Writing—review and editing; Kent C Sasse, Resources, Writing—review and editing, Revising the article for important clinical and intellectual content; Sean M Ward, Luis F Santana, Resources, Writing—review and editing; Yang K Xiang, Resources, Formal analysis, Writing—review and editing; Johannes W Hell, Resources, Formal analysis, Supervision, Writing—review and editing; Madeline Nieves-Cintrón, Conceptualization, Resources, Data curation, Formal analysis, Funding acquisition, Supervision, Investigation, Visualization, Methodology, Writing—original draft, Writing—review and editing; Manuel F Navedo, Conceptualization, Resources, Data curation, Formal analysis, Supervision, Funding acquisition, Validation, Investigation, Visualization, Methodology, Writing—original draft, Project administration, Writing—review and editing

## Author ORCIDs

Luis F Santana (iD) http://orcid.org/0000-0002-4297-8029
Manuel F Navedo (iD) http://orcid.org/0000-0001-6864-6594

## Ethics

Human subjects: This statement is already in our Materials and Methods section - Samples were obtained after Institutional Review Board (IRB) approval from the University of Nevada Reno School of Medicine (IRB ID: 2013-019) and in accordance with the guidelines of the Declaration of Helsinki. The need for informed consent was waived by IRBs at the University of Nevada Reno School of Medicine (IRB ID: 2013-019) and the University of California Davis (IRB ID: 597267-1) because the tissue is considered "waste", has no codification that could be used to identify patients and was determined not to be human subject research in accordance with United States of America federal regulations, as defined by 45 CFR 46.102(f). This precludes the acquisition of detailed clinical profiles other than

sex, age and whether the patient nondiabetic or diabetic. Only samples from nondiabetic patients were used.

Animal experimentation: This statement is already in our Materials and Methods section -Male wild-type C57BL/6J (wt) mice of 5 to 8 weeks of age were euthanized with a lethal dose of sodium pento-barbital (250 mg/kg; intraperitoneally), as approved by the University of California, Davis Animal Care and Use Committee (protocol #: 20321).

## Decision letter and Author response
Decision letter https://doi.org/10.7554/eLife.42214.073
Author response https://doi.org/10.7554/eLife.42214.074

## Additional files
### Supplementary files
• Supplementary file 1. K$^+$-induced constriction and baseline and passive diameters of arteries from wild type and S1928A mice.
DOI: https://doi.org/10.7554/eLife.42214.068

• Supplementary file 2. Arterial tone from wild type and S1928A mouse arteries.
DOI: https://doi.org/10.7554/eLife.42214.069

• Supplementary file 3. Human nondiabetic patients undergoing surgical sleeve gastrectomy.
DOI: https://doi.org/10.7554/eLife.42214.070

• Transparent reporting form
DOI: https://doi.org/10.7554/eLife.42214.071

### Data availability
All data generated or analyzed during this study are included in the manuscript as supporting files - source data files for each dataset.

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
