## [Decision Letter]

Thank you for submitting your article "A G_s_-coupled purinergic receptor boosts Ca^2+^ influx and vascular excitability during diabetic hyperglycemia" for consideration by *eLife*. Your article has been reviewed by three peer reviewers, including Richard S Lewis as the Reviewing Editor and Reviewer #1, and the evaluation has been overseen by Richard Aldrich as the Senior Editor. The following individual involved in review of your submission has agreed to reveal their identity: Mark T Nelson (Reviewer #2). A further reviewer remains anonymous.

The reviewers have discussed the reviews with one another and the Reviewing Editor has drafted this decision to help you prepare a revised submission.

Summary:

Cardiovascular complications of diabetes arise from augmented vasoconstriction of arterial myocytes in response to elevated glucose. In this manuscript, Navedo and colleagues demonstrate a pivotal role for the Gs- coupled metabotropic purinergic receptor, P2Y_11_, in mediating the vasoconstrictor response to glucose in mouse and human vascular smooth muscle myocytes. Using molecular, pharmacological/functional and FRET-based approaches, the authors provide evidence that P2Y_11_ is expressed in mouse and human vascular smooth muscle cells, and that glucose causes vasoconstriction through a cascade involving ATP release, P2Y_11_ activation, cAMP synthesis, and PKA-dependent phosphorylation of Ca_V_1.2. In support of the PKA-dependent activation of Ca_V_1.2 and subsequent vasoconstriction, they show using proximity analyses that a subpopulation of P2Y_11_ is in close proximity to PKA and Ca_V_1.2. The signaling specificity afforded by such microdomains may explain the paradoxical vasoconstriction-promoting effect of PKA, which is generally associated with vasodilatory responses through effects on K^+^ channels. The role of P2Y_11_ is convincing and is a major new insight. Also significant is the identification of smooth muscle P2Y_11_ as a potential target in the management of diabetic vasculopathy.

Essential revisions:

1) There were concerns about whether the glucose responses are affected by possible changes in osmolarity. From the Materials and methods, it does not appear that changes in osmolarity were controlled for, and the effect of glucose may be different in hypertonic and isotonic conditions. A series of controls for all experiments should be included in the manuscript rather than referencing an abstract (Syed et al., 2018).

2) Antibodies need to be validated and quantification described in more detail.a) For assessing Ser^1928^ phosphorylation, showing full-length blots with only a single molecular weight does not demonstrate specificity. The method of quantifying band densities should be described in more detail (more than just saying ImageJ was used). Were band densities integrated, and how were doublets quantified (e.g., Figures 1, 6, and especially Figure 7C)? In Figure 7C, MW markers are needed; is the upper Cav band of the doublet phosphorylated, and could this be used to estimate the proportion of channels that are phosphorylated? It is difficult to assess the linearity of the antibody without showing on the same blot dephosphorylated channels and fully phosphorylated channels. Does a 30% increase mean 30% of the channels are phosphorylated?

b) Primary antibodies to P2Y_11_ and PKA_cat_ need to be validated. It appears that P2Y_11_ ab is not specific for P2Y_11_ in mouse (extra bands in Figure 2—figure supplement 1A). This is important because the only evidence that P2Y_11_ is expressed in arterial myocytes comes from cell staining (Figure 2); the Westerns are from lysates also containing endothelial cells which are known to express P2Y_11_.

3) While ATP might be expected to accumulate in restricted spaces in tissue to activate P2Y_11_, there is some doubt about whether ATP release could be responsible for the increased Ba2+ currents in patch-clamp experiments where ATP would be free to diffuse. More evidence should be provided to support the role of ATP. For example, does continuous bath perfusion reduce or eliminate the increase in Ba2+ current in response to glucose? Can ATP release be detected by luciferase in the bath (e.g., see Ospichuk and Cahalan, Nature 359:241, 1992)? Finally, please discuss possible mechanisms of ATP secretion and what is known about the affinity of P2Y11 for ATP.

4) To show that phosphorylation of Ser^1928^ is essential for the regulation of contractility by P2Y_11_ agonists (and not merely correlated), vasoconstriction measurements (Figure 7) should be repeated with a S1928A knockin mouse. Similarly, the single-channel Po (Figure 8) should also be measured in the knockin animals to counter the possibility that prolonged incubation in glucose may have increased Po by affecting other signaling pathways and gene expression.

5) The specificity of LTCC modulation by glucose is an important issue, and the GSD data as they stand are rather descriptive and do not establish whether P2Y_11_ exists in complexes with AC, PKA, and LTCC. While it is beyond the scope of this paper to establish the composition of a P2Y_11_ signaling complex, the existence of such a complex would make the distribution of P2Y_11_, PKA, and Ca_V_1.2 non-random. To address this point, experimentally observed results should be compared with those of a simulated random distribution of the proteins at the same overall density, and the results shown in a supplementary figure.

6) Figures 4 and 5: The FRET response (cAMP production) is about twice as large in human myocytes than in mouse myocytes (normalized FRET ratio change, ~0.1 vs. ~0.05), despite the fact that the density of P2Y_11_-Ca_V_1.2 clusters in human myocytes is about 20% that in mouse myocytes (~0.03/µm2 vs. >0.15/µm2) and P2Y_11_-PKA cluster density is about the same in both humans and mice (~0.06/µm2). Why is cAMP production noticeably higher in humans than in mice, and is this difference significant? Please provide comment in the text.

7) A microdomain containing P2Y_11_, Ca_V_1.2, PKA and AC5 is discussed in the Discussion, but no data are shown for a role of AC5, only a citation of a FASEB meeting abstract (Discussion, first paragraph; Syed et al., 2018). Ideally, these data should be included in the current manuscript.

8) In the last paragraph of the subsection “The P2Y_11_ inhibitor NF340 prevents glucose-induced Ser^1928^ phosphorylation, LTCC activity and vasoconstriction”, the NF546 experiment does not rule out involvement of P2Y_1_ receptor, it merely shows it is not necessary for the vasoconstriction. To show it does not contribute, the authors should add the inhibitor after glucose to see if it inhibits (Figure 6—figure supplement 1D).

[Editors' note: further revisions were requested prior to acceptance, as described below.]

Thank you for resubmitting your work entitled "A G_s_-coupled purinergic receptor boosts Ca^2+^ influx and vascular excitability during diabetic hyperglycemia" for further consideration at *eLife*. Your revised article has been favorably evaluated by Richard Aldrich (Senior Editor) and a Reviewing Editor.

The manuscript has been improved and the reviewers' comments have been addressed well with new experimental data and additions to the text. However, several points need to be explained more clearly before the manuscript can be accepted, as outlined below:

Point 1. Please show a bar graph quantifying the pSer Western in Figure 1—figure supplement 1E (like that shown in Figure 1C) to make the point.

Point 2b. Please show two MW markers (55, 35) for the P2Y_11_ blot in Figure 2—figure supplement 1D since the band is well below 55 kD. This is needed to show expression at the expected MW of 40 kD.

Point 3. Please describe the perfusion experiment in more detail – in particular, the speed of perfusion (ml/min), and the time interval between changes in perfusion rate and collection of data. Please state the EC50 for stimulating P2Y_11_ with ATP in the text, with a reference.

Point 5. The simulations and analysis need to be explained in the Materials and methods section. How is "overlap" quantified in Figure 3I? It is surprising that almost no overlap is visible in the experimental data in Figure 3—figure supplement 3C ("overlap" panel). Briefly describe how was randomization done and how many times it was repeated. The explanation in the second paragraph of the subsection “P2Y_11_ receptors are in close proximity to Ca_V_1.2 and PKA_cat_ in arterial myocytes” does not adequately explain how the simulation was done, and what we are looking at in the images in Figure 3—figure supplement 3C.

---

## [Author Response]

Essential revisions:1) There were concerns about whether the glucose responses are affected by possible changes in osmolarity. From the Materials and methods, it does not appear that changes in osmolarity were controlled for, and the effect of glucose may be different in hypertonic and isotonic conditions. A series of controls for all experiments should be included in the manuscript rather than referencing an abstract (Syed et al., 2018).

To address the concerns regarding possible osmolarity effects, we performed experiments using equimolar concentrations of the nonpermeable mannitol (20 mM mannitol = 10 mM D-glu + 10 mM mannitol). Our new data show that elevated mannitol failed to induce vasoconstriction as well as to potentiate L-type Ca_V_1.2 channel (LTCC) activity, increase phosphorylation of Ca_V_1.2 at serine 1928 (Ser^1928^) or stimulate cAMP production in arterial myocytes. These results suggest that effects observed in cAMP synthesis, Ser^1928^ phosphorylation, LTCC activity and vascular reactivity in response to an increase in extracellular glucose (e.g. 20 mM D-glu) are not due to osmolarity effects but are rather a direct consequence of elevated glucose. These data are consistent with previous studies from our group and others (Jackson et al., 2016; Navedo et al., 2010; Nystoriak et al., 2014; Nystoriak et al., 2017; Rainbow et al., 2006). Moreover, the new data are in line with our recent observations indicating that the elevated glucose effects on LTCC activity and vascular reactivity require glucose internalization and metabolic utilization (Nystoriak et al., 2017). These new data and discussion are now included as Figure 1—figure supplement 1B, 1C and 1E (subsection “Glucose-induced extracellular nucleotide release mediates Ser^1928^ phosphorylation, increased LTCC activity and vasoconstriction”, first, second and third paragraphs), Figure 5—figure supplement 1B and 1C (subsection “Increased sarcolemmal cAMP synthesis in arterial myocytes in response to a P2Y_11_ agonist recapitulates glucose effects”, last paragraph) and Figure 8A and 8B (subsection “Increased Ser^1928^ phosphorylation and LTCC activity during chronic diabetic hyperglycemia are prevented by a P2Y_11_ inhibitor”).

2) Antibodies need to be validated and quantification described in more detail.a) For assessing Ser^1928^ phosphorylation, showing full-length blots with only a single molecular weight does not demonstrate specificity.

We appreciate this comment and apologize for not addressing the specificity of the antibodies clearly in our original manuscript. The specificity of the phospho-specific antibody for Ser^1928^ as well as the FP1 antibody for total Ca_V_1.2 measurement has been extensively and thoroughly validated in multiple published studies by our group, including for use in arterial myocytes (Buonarati et al., 2017; Hall et al., 2013; Nystoriak et al., 2017; Patriarchi et al., 2016). Importantly, immunoreactivity at 250 kDa of Ca_V_1.2 with the phospho-specific antibody for Ser^1928^ (but not a phospho-specific antibody for Ser^1700^) and FP1 are completely eliminated in tissue from S1928A KI mice (Figure 3 in Patriarchi et al., 2016) and conditional Ca_V_1.2 KO mice (Figure 2 in Buonarati et al., 2017), respectively. Full blots are shown in Buonarati et al., 2017, although the 100 kDa marker had been run to the bottom of the gel for better resolution of the high molecular mass Ca_V_1.2. Blotted membranes were trimmed to save on the use of these antibodies, which are generated in-house and only available in limited quantity. We have modified the original text to clarify this issue (subsection “Glucose-induced extracellular nucleotide release mediates Ser^1928^ phosphorylation, increased LTCC activity and vasoconstriction”, third paragraph; subsection “P2Y_11_ receptors are in close proximity to Ca_V_1.2 and PKA_cat_ in arterial myocytes”, second paragraph).

The method of quantifying band densities should be described in more detail (more than just saying ImageJ was used). Were band densities integrated, and how were doublets quantified (e.g., Figures 1, 6, and especially Figure 7C)?

We apologize for this oversight. We have revised the Materials and methods section of our manuscript to include an expanded description on quantification of Western blot data (see subsection “Immunoprecipitation and immunoblotting”).

In Figure 7C, MW markers are needed.

Thank you for pointing this out. We have added MW markers to all our blots in the main and supplementary figures.

Is the upper Cav band of the doublet phosphorylated, and could this be used to estimate the proportion of channels that are phosphorylated?

The upper band represents the full length form of Ca_V_1.2, which harbors Ser^1928^ (Hell et al., 1993). The lower band represents a shorter Ca_V_1.2 form that is truncated around residue 1800 and thus does not contain Ser^1928^. Accordingly, only the upper band can be phosphorylated on Ser^1928^. The shorter channel form still has the proteolytically cleaved C-terminal domain attached to it, but this C-terminal fragment is difficult to detect with the small amounts of material from our arterial samples.

To define basal Ser^1928^ phosphorylation levels, we previously used Ca_V_1.2 isolated from rat forebrain where we obtain much more material than from our arterial samples. There, we estimated that ~16.7% of the Ca_V_1.2 long form is phosphorylated on Ser^1928^ under basal conditions in healthy adult rats (Davare and Hell, 2003). However, this is only an upper limit. With our arterial preparation (freshly isolated small resistance arteries from cerebral and mesenteric vessels) and the small amount of working material obtained from it, it could be less than that because the only way we can obtain such an estimate is by comparing the immunoblot signal from Ca_V_1.2 from untreated samples with those where Ca_V_1.2 had been incubated after the immunoprecipitation with purified PKA and Mg-ATP. The estimate assumes that all full-length Ca_V_1.2 is phosphorylated on Ser^1928^ under these stimulatory conditions, but unfortunately, we cannot be completely certain about this. Hence, we will not be able to determine with high confidence the percentage of total Ca_V_1.2 that is phosphorylated at Ser^1928^ under basal conditions in our preparation. Such a number would also not necessarily be too helpful because a substantial fraction of Ca_V_1.2 resides inside the cells (Conrad et al., 2018; Ghosh et al., 2018) when only channels at the plasma membrane are relevant for potentiation by Ser^1928^ phosphorylation. For instance, if we would find that ~16% of Ca_V_1.2 is phosphorylated in arterial lysates under basal conditions, it could quite well reflect that 50% of Ca_V_1.2 at the surface membrane are Ser^1928^ phosphorylated if the intracellular pool constitutes 70% of total Ca_V_1.2 and is largely not phosphorylated. We will be working in future studies to address this interesting point in a rigorous manner but would like to respectfully note that these data will not change the conclusions of the present study.

It is difficult to assess the linearity of the antibody without showing on the same blot dephosphorylated channels and fully phosphorylated channels.

We show dephosphorylated and phosphorylated channels on the same blot – e.g. total Ca_V_1.2 reprobed after Ser^1928^ phosphorylation probe. Moreover, our blots are analyzed after films are exposed for increasing time periods (1 min, 2 min, 4 min) to ensure that the signals are in the linear range, as detailed in our previous work (Davare and Hell, 2003; Hall et al., 2006). This information is clarified and expanded in our revised manuscript (subsection “Immunoprecipitation and immunoblotting”).

Does a 30% increase mean 30% of the channels are phosphorylated?

No. It means the percentage of channels that were phosphorylated has increased by 30%. As an example, if 10% of the channels are phosphorylated basally and treatment increases the phosphorylation state by 30%, then ~13% of the channels would be phosphorylated upon treatment.

b) Primary antibodies to P2Y_11_ and PKA_cat_ need to be validated. It appears that P2Y_11_ ab is not specific for P2Y_11_ in mouse (extra bands in Figure 2—figure supplement 1A). This is important because the only evidence that P2Y_11_ is expressed in arterial myocytes comes from cell staining (Figure 2); the Westerns are from lysates also containing endothelial cells which are known to express P2Y_11_.

To address this question, we performed additional experiments. The PKA_cat_ Ab was validated using a blocking peptide provided by the vendor (Santa Cruz Biotechnology). Briefly, immunofluorescent labeling of PKA_cat_ in arterial myocytes was performed using a mouse anti-PKA_cat_ Ab (Santa Cruz Biotechnology) incubated with or without a PKA_cat_ blocking peptide (A-2; Santa Cruz Biotechnology) before applying to cells on coverslips. Consistent with a previous study from our group (Nystoriak et al., 2017), arterial myocytes showed broad intracellular PKA_cat_-associated fluorescence, which was completely lost when the PKA_cat_ Ab was preabsorbed with the PKA_cat_ blocking peptide. These data are now presented in Figure 3—figure supplement 1 (subsection “P2Y_11_ receptors are in close proximity to Ca_V_1.2 and PKA_cat_ in arterial myocytes”, second paragraph) of the revised manuscript.

As for the P2Y_11_ Ab validation, we performed multiple tests. First, we detected single protein bands of expected molecular weight in lysates from untransfected and vehicle treated tsA-201 cells, which are known to endogenously express P2Y_11_ (Dreisig and Kornum, 2016). When tsA-201cells were transfected with a vector encoding a P2Y_11_ tagged with GFP (P2Y_11_-GFP), two bands were detected. The small molecular weight band corresponds to the endogenous P2Y_11_ and the other band is consistent with the addition of GFP to P2Y_11_ (see Figure 2A). Treatment of tsA-201 cells over-expressing P2Y_11_-GFP with specific P2Y_11_ antisense oligodeoxynucleotides (ODNs) resulted in a reduction in the intensity of both bands (Figure 2B). In native arterial tissue, we detected an immunoreactive band of the expected molecular weight for P2Y_11_ (~40 kDa) with the same P2Y_11_ Ab (Figure 2C and Figure 2—figure supplement 1A). Importantly, the abundance of this ~40 kDa band was reduced to about ~50% of the basal P2Y_11_ protein abundance in arteries treated with specific P2Y_11_ antisense ODNs (Figure 2—figure supplement 1B). A band of ~110-120 kDa was observed in mouse arterial lysates, perhaps reflecting the formation of heterodimers between the P2Y_11_ receptor itself or other purinergic/ G protein-coupled receptors, as previously reported (Barragan-Iglesias et al., 2015; Barragan-Iglesias et al., 2014; Ecke et al., 2008; Nishimura et al., 2016). This discussion is included in the first paragraph of the subsection “P2Y_11_ receptors are in close proximity to Ca_V_1.2 and PKA_cat_ in arterial myocytes” We also observed a band of expected molecular weight for P2Y_11_ in lysates from isolated mouse arterial myocytes (Figure 2—figure supplement 1D), and P2Y_11_-associated fluorescence was not detected in isolated mouse arterial myocytes exposed to a boiled P2Y_11_ Ab (Figure 2—figure supplement 1E). In addition, note that the functional expression of human P2Y_11_ and mouse P2Y_11_-like receptors in arterial myocytes is further supported by multiple well-controlled functional studies described throughout the manuscript (please see Figures 5-8). Finally, we reached out to several commercial companies that sell P2Y_11_ Ab to obtain antigenic peptides for P2Y_11_. They do not have available antigenic peptides for P2Y_11_ Ab nor were able to provide us with the amino acid sequence used to raise the antibody, thus prohibiting that additional approach to antibody validation. Regardless, we consider that our multiple biochemical and functional approaches provide strong support for validation of the P2Y_11_ Ab and the presence of human P2Y_11_ and mouse P2Y_11_-like receptor in arterial myocytes.

3) While ATP might be expected to accumulate in restricted spaces in tissue to activate P2Y_11_, there is some doubt about whether ATP release could be responsible for the increased Ba2+ currents in patch-clamp experiments where ATP would be free to diffuse. More evidence should be provided to support the role of ATP. For example, does continuous bath perfusion reduce or eliminate the increase in Ba2+ current in response to glucose? Can ATP release be detected by luciferase in the bath (e.g., see Ospichuk and Cahalan, Nature 359:241, 1992)? Finally, please discuss possible mechanisms of ATP secretion and what is known about the affinity of P2Y11 for ATP.

We recognize that our original cartoon identifying extracellular ATP as a primary mediator of glucose-induced activation of P2Y_11_ was premature. Although we are now including additional data showing that the well-known non-hydrolyzed ATP analog and P2Y_11_ agonist ATPγS (von Kugelgen and Harden, 2011) stimulates Ba^2+^ currents in arterial myocytes (Figure 7—figure supplement 1B; subsection “The P2Y_11_ agonist NF546 stimulates Ser^1928^ phosphorylation, PKA-dependent LTCC activity and vasoconstriction”, first paragraph), we have modified the illustration to avoid data overinterpretation.

Note, however, that increased glucose (20 mM D-glucose) has been shown to trigger the release of NUC into the extracellular space of many cells, including arterial myocytes (Costa et al., 2009; Hazama et al., 1998; Nilsson et al., 2006; Parodi et al., 2002). Our results with apyrase support this conclusion (Figure 1). In addition, we now provide additional data to support the concept that glucose-induced extracellular nucleotide release mediates the changes in Ba^2+^ current in isolated voltage clamped arterial myocytes by performing the proposed experiments with continuous bath perfusion. Accordingly, we were able to record robust Ba^2+^ current in isolated voltage clamped arterial myocytes under static and flow conditions. However, as predicted by the reviewers, whereas I_Ba_ recorded in a static bath was significantly elevated upon increasing external D-glucose from 10 mM to 20 mM, this response was not observed under continuous flow conditions. These results suggest that nucleotides (NUC) could indeed diffuse away under continuous flow conditions, whereas they could accumulate in or around the area of release at the surface membrane of arterial myocytes to induce the optimal activation of the signaling pathway underlying the increased Ba^2+^ current density in response to 20 mM D-glucose. We did attempt to detect glucose-induced NUC release (e.g. ATP) using a commercially available luciferin-luciferase kit. Unfortunately, we could not detect any signal. After consulting with the manufacturer, this was attributed to a need for optimizing the preparation to detect NUC release from native arterial myocytes. This is not a trivial task and may take more time than that allotted for resubmission of the manuscript. We are currently working to optimize our preparation/assay and rigorously address this point in a comprehensive follow up study. However, new data provide further support to our hypothesis that elevated glucose evokes LTCC activity and vasoconstriction through an autocrine mechanism involving secreted nucleotides, which are well-known activators of G-coupled P2Y receptors (von Kugelgen and Harden, 2011). These new data and discussion are now included as Figure 1—figure supplement 3 (subsection “Glucose-induced extracellular nucleotide release mediates Ser^1928^ phosphorylation, increased LTCC activity and vasoconstriction”).

As suggested, we provide a discussion of possible mechanisms of NUC (including ATP) secretion and P2Y_11_ pharmacological profile (see Discussion, sixth paragraph).

4) To show that phosphorylation of Ser^1928^ is essential for the regulation of contractility by P2Y_11_ agonists (and not merely correlated), vasoconstriction measurements (Figure 7) should be repeated with a S1928A knockin mouse. Similarly, the single-channel Po (Figure 8) should also be measured in the knockin animals to counter the possibility that prolonged incubation in glucose may have increased Po by affecting other signaling pathways and gene expression.

To address this question, we performed additional experiments. First, we found that arteries from S1928A mice failed to constrict in response to the P2Y_11_ agonist NF546. These results are in stark contrast to those observed for wild type arteries (see Figure 7E and 7F), and further support the conclusion that phosphorylation of Ser^1928^ is essential for the regulation of vascular reactivity in response to a P2Y_11_ agonist. These new data and discussion are now included as Figure 7F in the revised manuscript (subsection “The P2Y_11_ agonist NF546 stimulates Ser^1928^ phosphorylation, PKA-dependent LTCC activity and vasoconstriction”, last paragraph).

Second, we performed the suggested single-channel experiments in arterial myocytes from S1928A mice under chronically elevated glucose stimulation (48 hours). Results indicate that S1928A cells isolated from either 10 mM and 20 mM D-glucose groups had similar LTCC open probability (nP_o_), which are comparable to that observed for LTCCs in wild type cells isolated from the group treated with 10 mM D-glucose. These results counter the possibility that prolonged incubation in elevated glucose may have increased LTCC nP_o_ by affecting other signaling pathways and gene expression, and further support a key role for Ser^1928^ phosphorylation in regulation of LTCC activity during diabetic hyperglycemia. These new data and discussion are now included as Figure 8—figure supplement 1B (subsection “Increased Ser^1928^ phosphorylation and LTCC activity during chronic diabetic hyperglycemia are prevented by a P2Y_11_ inhibitor”) in the revised manuscript.

5) The specificity of LTCC modulation by glucose is an important issue, and the GSD data as they stand are rather descriptive and do not establish whether P2Y_11_ exists in complexes with AC, PKA, and LTCC. While it is beyond the scope of this paper to establish the composition of a P2Y_11_ signaling complex, the existence of such a complex would make the distribution of P2Y_11_, PKA, and Ca_V_1.2 non-random. To address this point, experimentally observed results should be compared with those of a simulated random distribution of the proteins at the same overall density, and the results shown in a supplementary figure.

To address this point, we performed the suggested comparison between experimental and simulated random protein distribution. Accordingly, we compared the frequency of P2Y_11_-Ca_V_1.2 and P2Y_11_-PKA overlap that was obtained experimentally with that from simulated images that were randomized for proteins of interest at the same overall density as those observed in the experimental super-resolution maps (Bolte and Cordelieres, 2006). The analysis revealed a higher level of overlap between P2Y_11_-Ca_V_1.2 and P2Y_11_-PKA in the experimental data than that observed for a simulated random distribution. These results suggest a selective and distinctive interaction between a subpopulation of P2Y_11_ and a pool of Ca_V_1.2 and PKA, which complements our super-resolution data in the first submission. These new data and discussion are now included as Figure 3I and Figure 3—figure supplement 3C (subsection “P2Y_11_ receptors are in close proximity to Ca_V_1.2 and PKAcat in arterial myocytes”, second paragraph) in the revised manuscript.

6) Figures 4 and 5: The FRET response (cAMP production) is about twice as large in human myocytes than in mouse myocytes (normalized FRET ratio change, ~0.1 vs. ~0.05), despite the fact that the density of P2Y_11_-Ca_V_1.2 clusters in human myocytes is about 20% that in mouse myocytes (~0.03/µm2 vs. >0.15/µm2) and P2Y_11_-PKA cluster density is about the same in both humans and mice (~0.06/µm2). Why is cAMP production noticeably higher in humans than in mice, and is this difference significant? Please provide comment in the text.

To answer this question, we first respectfully submit that it would not be appropriate at this time to correlate PLA data with the FRET responses as many factors may independently influence each measurement. The PLA data indicate relative abundance of the protein complex whereas the FRET data indicate relative signaling intensity after stimulation. There is not a necessary correlation between these two results. Accordingly, the goal of our PLA experiments is to define close association between P2Y_11_-Ca_V_1.2 and P2Y_11_-PKA, whereas the goal of the FRET experiments is to determine whether glucose could modulate cAMP signaling as well as define potential mechanisms such as the involvement of the P2Y_11_. We strongly believe that our data have accomplished these goals.

The cAMP production in response to glucose and the P2Y_11_ agonist NF546 is indeed significantly larger in human arterial myocytes compared to mouse cells (*P* < 0.05). The reasons for this are currently unclear but may include intrinsic differences associated with species (human vs. mouse), blood vessels (human adipose arteries vs mouse aortae) and expression patterns of the biosensor in human vs. mouse cells, as well as variations in protein expression and receptor/enzyme activity (e.g. P2Y_11_, AC and PDE expression/activity) in human vs. mouse arterial myocytes. Whether either or all of the factors described above contribute to distinctive cAMP responses in human vs mouse arterial myocytes as well as their physiological implications, remain to be explored. Nevertheless, as suggested, we now provide a description of these issues in the revised manuscript (subsection “Increased sarcolemmal cAMP synthesis in arterial myocytes in response to a P2Y_11_ agonist recapitulates glucose effects”, last paragraph).

7) A microdomain containing P2Y_11_, Ca_V_1.2, PKA and AC5 is discussed in the Discussion, but no data are shown for a role of AC5, only a citation of a FASEB meeting abstract (Discussion, first paragraph; Syed et al., 2018). Ideally, these data should be included in the current manuscript.

We recognize that our discussion of AC5 in complex withP2Y_11_, Ca_V_1.2 and PKA is premature. We have therefore modified the text as reference to AC5 does not directly relate to the focus of this manuscript. We are working to rigorously and comprehensively examine the role of AC5 in a follow up manuscript.

8) In the last paragraph of the subsection “The P2Y_11_ inhibitor NF340 prevents glucose-induced Ser^1928^ phosphorylation, LTCC activity and vasoconstriction”, the NF546 experiment does not rule out involvement of P2Y_1_ receptor, it merely shows it is not necessary for the vasoconstriction. To show it does not contribute, the authors should add the inhibitor after glucose to see if it inhibits (Figure 6—figure supplement 1D).

To address this question, we performed additional experiments. We now show that glucose-mediated vasoconstriction is not affected by adding the P2Y_1_ inhibitor MRS2179 either before or after 20 mM D-glucose. These new data and discussion are now included as Figure 6—figure supplement 1E (subsection “The P2Y_11_ inhibitor NF340 prevents glucose-induced Ser^1928^ phosphorylation, LTCC activity and vasoconstriction”, last paragraph) in the revised manuscript.

[Editors' note: further revisions were requested prior to acceptance, as described below.]

The manuscript has been improved and the reviewers' comments have been addressed well with new experimental data and additions to the text. However, several points need to be explained more clearly before the manuscript can be accepted, as outlined below:Point 1. Please show a bar graph quantifying the pSer Western in Figure 1—figure supplement 1E (like that shown in Figure 1C) to make the point.

A bar plot quantifying pSer^1928^ in the presence of 10 mM D-glucose and 20 mM mannitol is now included in the revised version of the manuscript (Figure 1—figure supplement 1E).

Point 2b. Please show two MW markers (55, 35) for the P2Y_11_ blot in Figure 2—figure supplement 1D since the band is well below 55 kD. This is needed to show expression at the expected MW of 40 kD.

As requested, we now show two MW marker for the WB blot in Figure 2—figure supplement 2D.

Point 3. Please describe the perfusion experiment in more detail – in particular, the speed of perfusion (ml/min), and the time interval between changes in perfusion rate and collection of data. Please state the EC_50_ for stimulating P2Y_11_ with ATP in the text, with a reference.

We have expanded our description of the perfusion experiments and have provided additional information about time intervals between perfusion rate and collection of data in the Materials and methods section and in the figure legend of the revised manuscript (subsection “Electrophysiology”, first paragraph and Figure 1—figure supplement 2 legend). We also stated the EC_50_ for stimulating P2Y_11_ with ATP and have provided citations for this in the revised manuscript (subsection “The P2Y_11_ agonist NF546 stimulates Ser^1928^ phosphorylation, PKA-dependent LTCC activity and vasoconstriction”, first paragraph, and Discussion, sixth paragraph).

Point 5. The simulations and analysis need to be explained in the Materials and methods section. How is "overlap" quantified in Figure 3I? It is surprising that almost no overlap is visible in the experimental data in Figure 3—figure supplement 3C ("overlap" panel). Briefly describe how was randomization done and how many times it was repeated. The explanation in the second paragraph of the subsection “P2Y_11_ receptors are in close proximity to Ca_V_1.2 and PKA_cat_ in arterial myocytes” does not adequately explain how the simulation was done, and what we are looking at in the images in Figure 3—figure supplement 3C.

To address these points, we have expanded the description of the generation of simulation/randomization data as well as the overall analysis in the revised manuscript. Randomization of P2Y_11_-Ca_V_1.2 and P2Y_11_-PKA_cat_ was repeated six times and then compared against experimental data obtained from the super-resolution localization maps. The percentage of P2Y_11_ overlap was determined by multiplying thresholded, binary localization maps (experimental and simulated data) of P2Y_11_by those from Ca_V_1.2 or PKA_cat_. We apologize and acknowledge that the quality of original images in Figure 3—figure supplement 3C was not adequate, thus preventing proper evaluation of the data. We have modified the presentation of these images by changing the LUT to an inverted grayscale LUT that makes the overlapping signals more apparent and therefore, better reflects the results from the amalgamated data. Changes are found in subsection “P2Y_11_ receptors are in close proximity to Ca_V_1.2 and PKA_cat_ in arterial myocytes”, second paragraph, subsection “Immunolabeling and Ground State Depletion (GSD) microscopy”, last paragraph, and in the Figure 3—figure supplement 3 legend.